# Structural basis for HIV-1 capsid adaption to a deficiency in IP6 packaging

Yanan Zhu [1,2,7], Alex B. Kleinpeter [3,7], Juan S. Rey [4,7], Juan Shen[1], Yao Shen [1], Jialu Xu[1], Nathan Hardenbrook[1], Long Chen [1], Anka Lucic[1], Juan R. Perilla [4] ✉, Eric O. Freed [3] ✉ & Peijun Zhang [1,5,6] ✉

Inositol hexakisphosphate (IP6) promotes HIV-1 assembly by stabilizing the immature Gag lattice and becomes enriched within virions, where it is required for mature capsid assembly. Previously, we identified Gag mutants that package little IP6 yet assemble particles, though they are non-infectious due to defective capsid formation. Here, we report a compensatory mutation, G225R, in the C-terminus of capsid protein (CA) that restores capsid assembly and infectivity in these IP6-deficient mutants. G225R also enhances in vitro assembly of CA into capsid-like particles at far lower IP6 concentrations than required for wild-type CA. CryoEM structures of G225R CA hexamers and lattices at 2.7 Å resolution reveal that the otherwise disordered C-terminus becomes structured, stabilizing hexamer-hexamer interfaces. Molecular dynamics simulations support this mechanism. These findings uncover how HIV-1 can adapt to IP6 deficiency and highlight a previously unrecognized structural role of the CA C-terminus, while offering tools for capsid-related studies.

Nascent HIV-1 virions assemble at the plasma membrane of infected cells, packaging both viral and host-cell components required for the next round of infection. HIV-1 virion assembly is driven by the multimerization of the Gag precursor, a 55-kDa polyprotein (Pr55Gag) comprising matrix (MA), capsid (CA), nucleocapsid (NC), and p6 domains and two spacer peptides, SP1 and SP2, which connect CA-NC and NC-p6, respectively[1]. Gag oligomerizes into a spherical lattice of Gag hexamers—the immature Gag lattice—ultimately assembling into nascent particles that undergo membrane scission and release. Viral enzymes are recruited into growing particles via co-assembly of Gag with the GagPol polyprotein precursor, which is synthesized at a GagPol: Gag ratio of -1:20. GagPol contains the protease (PR), reverse transcriptase (RT), and integrase (IN) domains that are subsequently required for particle infectivity. Concomitant with their release from the infected cell, virions undergo maturation during which immature particles are morphologically transformed into mature virions containing a cone-shaped viral core. Maturation is initiated when PR cleaves specific sites in Gag and GagPol, liberating their constituent domains. CA then assembles into the mature capsid, a conical structure that encloses the viral RNA genome and enzymes RT and IN and serves as the protective shell of the viral core. The capsid is composed of 200–250 CA hexamers and exactly 12 CA pentamers, which are concentrated at each end to ensure capsid closure[2]. In the next round of infection, the core is deposited into the cytosol during virus entry and remains intact, ultimately passing through a nuclear pore and uncoating near sites of integration in the nucleus. Importantly, the assembly of a stable capsid during maturation, and its stability postentry, are crucial for reverse transcription of the HIV-1 genome, protection of the viral genome from innate immune sensing, trafficking of the viral core to the nuclear envelope, nuclear entry of the core

[1]Division of Structural Biology, Wellcome Centre for Human Genetics, University of Oxford, Oxford, UK. [2]Institute for Advanced Study in Physics, Zhejiang University, Hangzhou, Zhejiang, China. [3]Virus-Cell Interaction Section, HIV Dynamics and Replication Program, Center for Cancer Research, National Cancer Institute, Frederick, MD, USA. [4]Department of Chemistry and Biochemistry, University of Delaware, Newark, DE, USA. [5]Diamond Light Source, Harwell Science and Innovation Campus, Didcot, UK. [6]Chinese Academy of Medical Sciences Oxford Institute, University of Oxford, Oxford, UK. [7]These authors contributed equally: Yanan Zhu, Alex B. Kleinpeter, Juan S. Rey. ✉e-mail: jperilla@udel.edu; efreed@nih.gov; peijun.zhang@strubi.ox.ac.uk

through nuclear pores, and efficient integration of the newly synthe-sized viral DNA into gene-rich regions of the host genome[3].

The polyanionic host-cell metabolite inositol hexakisphosphate (IP6) is a key co-factor for HIV-1 capsid assembly and stability[4,5]. IP6 is coordinated by two positively charged amino acids in the CA N-terminal domain (CA-NTD)– Arg-18 (R18) and Lys-25 (K25)–that form electropositive rings within a pore at the center of CA hexamers and pentamers[4,6,7]. Mutation of R18 or K25, each of which can bind a molecule of IP6, drastically reduces capsid formation in virions and particle infectivity and prevents the efficient in vitro assembly of mature CA into capsid-like particles (CLPs) in the presence of IP6[4,6,8–12]. Several studies suggest that IP6 binding to R18 and K25 rings is parti-cularly crucial for the formation of CA pentamers, in which these rings are more tightly packed than in hexamers, and therefore more prone to destabilization via charge repulsion in the absence of IP6[7,10,11]. Because capsid formation occurs after particle release from a virus-producing cell and prior to fusion with a target cell, CA must rely on IP6 acquired from the virus-producing cell during particle assembly.

HIV-1 recruits IP6 during immature Gag lattice assembly via two Lys residues in the C-terminal domain of CA–Lys–158 (K158) and Lys–227 (K227)–which form positively charged rings at the center of immature Gag hexamers[4,13,14]. The final eight amino acids of CA (including K227) and SP1 participate in key intra-hexamer interactions stabilizing Gag hexamers and the immature Gag lattice by folding into a six-helix bundle (6HB)[14–16]. Mutations that destabilize the CA-SP1 6HB result in drastic defects in virus particle assembly, morphogenesis, and infectivity[17,18]. Conversely, because the CA-SP1 cleavage site is located inside the 6HB, mutations that hyper-stabilize the 6HB, such as SP1-T8I, obstruct PR access to this cleavage site, preventing CA-SP1 processing and reducing virion infectivity[17–19]. The stability of the CA-SP1 6HB is thus finely tuned to allow both efficient assembly and subsequent maturation.

IP6 plays a crucial role in 6HB stability by neutralizing the highly charged pore at the top of the 6HB formed by the K158 and K227 rings. Depletion of IP6 in virus-producing cells, or mutation of either K158 or K227, results in assembly defects caused by destabilization of the 6HB and reduced IP6 incorporation into viral particles[13,18,20]. This reliance on IP6 for efficient particle assembly ensures that IP6 is enriched in virions to promote the formation of the capsid during maturation. Our recent study showed that viruses harboring mutations at both Lys rings (K158A/K227A or "KAKA") in the immature Gag lattice could assemble independently of IP6[21]. Structural analysis revealed that KAKA virions contained an immature lattice that was largely indistinguishable from that in immature wild-type (WT) virions, apart from the absence of IP6 density[20,21]. However, KAKA virions were not infectious due to an inability to build stable capsids during particle maturation[21]. The above findings support a model in which the requirement for IP6 during particle assembly is a mechanism to enrich IP6 in virions to promote capsid assembly and subsequent infection.

Here we utilize KAKA and KAKA/T8I mutant viruses, which fail to bind and recruit IP6 during assembly, to further delineate the separate but interdependent roles of IP6 in particle assembly and maturation. After characterizing the effect of mutations in specific IP6-binding residues in authentic immature particles, we propagate these mutants to assess how HIV-1 could adapt to an inability to enrich particles with IP6. Propagation of KAKA and KAKA/T8I mutants lead to the acquisi-tion of a compensatory mutation, G225R, in the CA C-terminus that rescues KAKA/T8I conical capsid formation and particle infectivity without restoring IP6 enrichment during assembly. We determine the cryoEM structure of CA-KAKA/G225R CLPs at a 2.7 Å resolution, which reveal the structural mechanism by which G225R promotes mature HIV-1 capsid assembly in virions not enriched with IP6. Remarkably, we show that the G225R mutation results in an additional intermolecular interaction via the CA C-terminus, which is otherwise flexible and has thus far been refractory to structural analysis. We further show that the

G225R mutation promotes the in vitro assembly of CLPs in low-IP6 conditions. Our results suggest that the C-terminus of CA may play a heretofore unappreciated role in the assembly of the HIV-1 capsid and the modulation of capsid stability.

## Results
### CA-G225R rescues the replication defect imposed by IP6-packaging deficiency

Cryo-electron tomography (cryoET) combined with subtomogram averaging (STA) is a cutting-edge imaging technique that reveals the 3D structure of biological specimens at near-atomic resolution in their native, frozen-hydrated state. In previous studies, we used cryoET STA to investigate the effect of mutations, namely KAKA, in the IP6 binding pocket of immature Gag hexamers in the context of the WT and 6HB-stabilizing SP1 mutations T8I and M4L/T8I[20,21]. We found that immature KAKA VLPs were composed of Gag hexamers that were largely indis-tinguishable from those formed by WT Gag, apart from the absence of IP6 density at the top of the 6HB. To dissect further the contributions of individual single or dual mutations on the immature Gag lattice, we analysed the structures of immature K227A and KAKA/T8I VLPs pro-duced from 293T cells and compared them to the previously reported structures of WT and KAKA immature VLPs (Fig. 1). Both K227A and KAKA/T8I assemble into immature spherical VLPs similar to those formed by WT and KAKA (Fig. 1a–d). Our cryoET STA structure of K227A at 3.82 Å resolution showed an intermediate level of IP6 com-pared with the WT (full) and the KAKA mutant (empty) (Fig. 1e–g, Supplementary Figs. 1a, 2, Supplementary Table 1), consistent with the previous biochemical measurements[21]. The structure of KAKA/T8I at 3.67 Å resolution displayed no IP6 density (Fig. 1h, Supplementary Fig. 1b, Supplementary Table 1), similar to the KAKA mutant, sug-gesting that the 6HB-stabilizing mutation SP1-T8I does not restore IP6 enrichment to KAKA. Consistent with this inability to restore IP6 enrichment, the addition of SP1-T8I does not substantially reverse the infectivity defect imposed by the KAKA mutations[21] (Fig. 1i).

Next, we sought to determine how HIV-1 might adapt to an inability to enrich IP6 during particle assembly. We propagated KAKA and KAKA/T8I in the highly permissive MT-4 T cell line. Briefly, MT-4 cells were initially transfected with the WT NL4-3 infectious molecular clone or derivatives harboring the KAKA or KAKA/T8I mutations. Consistent with the drastic decrease in KAKA and KAKA/T8I infectivity, each mutant demonstrated a significant delay in replication compared to WT virus (Fig. 1j). Upon propagation of virus collected from the peak of replication, both KAKA and KAKA/T8I replicated with a substantially reduced delay relative to WT (Fig. 1k). This observation is suggestive of the acquisition of mutations that compensate for the defects induced by the KAKA and KAKA/T8I mutations. To confirm the acquisition of compensatory mutations, we isolated genomic DNA from infected cells at the peak of replication and PCR amplified the Gag coding region. Gag amplicons were sequenced, and potential compensatory mutations were cloned into the KAKA and KAKA/T8I NL4-3 molecular clones. This process was performed iteratively using mutant viruses selected in initial experiments to initiate downstream propagation experiments. We also performed these in vitro selection experiments in the C8166 T cell line and with additional mutant viruses containing a K158T mutation in place of K158A (KTKA) and/or an SP1-M4L mutation in place of T8I. Compensatory mutations selected in these experi-ments were found throughout CA and SP1 (Supplementary Table 2). Interestingly, several selected mutations were identified at amino acid positions associated with resistance to capsid inhibitors (e.g., CA-Thr-107, His-87, Ala-105, Thr-58, Gly-208, Thr-216)[22–25], altered capsid assembly and stability (e.g., CA-H12Y, N21S, T216I)[11,24,26], and host factor dependence (e.g., CA-Ala-77, His-87, Gly-94D, Gly-208, Thr-210, Glu-187, Pro-207)[27–30]. We also observed mutations at amino acid positions located at critical interfaces within the mature CA lattice including the β-hairpin (Val-11 and His-12), the central pore (Asn-21 and Ala-22), the

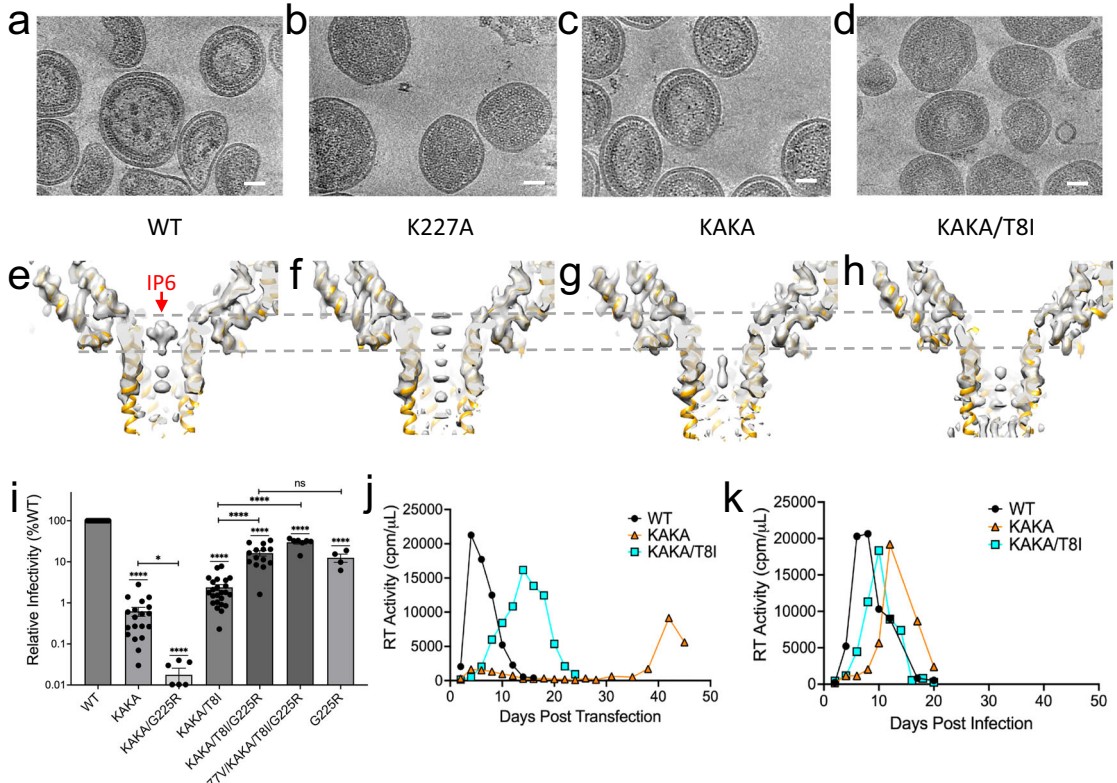

**Fig. 1 | Characterization of Gag mutants defective in IP6 binding and a compensatory mutation G225R. a–d** Representative tomographic slices of WT, K227A, K158A/K227A (KAKA) and KAKA/T8I VLPs, respectively. Scale bar, 50 nm. **e–h** CryoET STA structures of immature Gag hexamers from WT, K227A, KAKA and KAKA/T8I VLPs superimposed with the model (PDB: 7ASH). Dashed gray lines mark the height position for IP6. **i** Specific infectivity of KAKA and KAKA/T8I in the presence and absence of G225R measured in TZM-bl cells at 36–48 h post-infection. Data are the mean of at least three independent biological replicates for each mutant, and error bars depict ± SEM. Precise *n* for each group: WT = 24, KAKA = 19, KAKA/G225R = 6, KAKA/T8I = 24, KAKA/T8I/G225R = 14, A77V/KAKA/T8I/G225R = 7, G225R = 4. Replicates that did not produce a measurable signal were assigned a value of 0 and are plotted at the base on the y-axis for visualization. Statistical analysis was performed using GraphPad Prism. Statistical significance

was determined by a two-tailed one-sample Student's *t* test with a hypothetical value set to 100 when comparing groups to WT. All other comparisons were made using two-tailed unpaired Student's *t* tests. (*p*-value summary: >0.05 = not significant; <0.05 = *; <0.01 = **; <0.001 = ***; <0.0001 = ****). Precise *p*-values for each comparison: WT vs. KAKA $P < 0.0001$, WT vs. KAKA/G225R $P < 0.0001$, WT vs. KAKA/T8I $P < 0.0001$, WT vs. KAKA/T8I/G225R $P < 0.0001$, WT vs. A77V/KAKA/T8I/G225R $P < 0.0001$, WT vs. G225R $P < 0.0001$, KAKA vs KAKA/G225R $P = 0.0390$, KAKA/T8I vs KAKA/T8I/G225R $P < 0.0001$, KAKA/T8I vs A77V/KAKA/T8I/G225R $P < 0.0001$, KAKA/T8I/G225R vs G225R $P = 0.4633$. **j** Representative replication kinetics of KAKA and KAKA/T8I mutants in MT-4 cells showing a delay in viral replication relative to WT. **k** Representative re-passage of KAKA and KAKA/T8I in fresh MT-4 cells showing an increase in replication kinetics relative to the initial passage. Source data are provided as a Source Data file.

Thr-Val-Gly-Gly (TVGG) motif that controls CA hexamer/pentamer assembly (Thr-58, Gly-61), the NTD/CTD interface between adjacent CA protomers (Met-68, Ala-105, Thr-107), and the trimer interface between neighboring CA hexamers/pentamers (Pro-207, Gly-208, Thr-210, Thr-216).

To determine whether the mutations selected in these experiments could restore fitness to KAKA and KAKA/T8I, we quantified the effect of these mutations on the single-cycle infectivity of KAKA and KAKA/T8I. Most of the mutations that we observed were unable to significantly restore infectivity to KAKA or KAKA/T8I (Supplementary Fig. 3), including two mutations–N21S and T216I–that were recently reported to restore infectivity to K25A, a mutant unable to bind IP6 in the context of the mature CA lattice[11]. However, we did observe a mutation at position 225 in CA that significantly restored infectivity to KAKA/T8I. Initially, we observed a Gly-to-Ser substitution at residue 225 upon propagation of KAKA in MT-4 cells. After introducing the G225S mutation into multiple KAKA- and KTKA-containing clones, we observed a Ser-to-Arg substitution at CA position 225 (G225R) upon propagation of KTKA/T8I/G225S. We then introduced the G225R mutation into KAKA and KAKA/T8I to determine its effect on particle infectivity. We found that G225R conferred an ~8-fold increase in

single-cycle infectivity to KAKA/T8I (from ~2% of WT to 16%) (Fig. 1i). Further propagation of KAKA/T8I/G225R resulted in the acquisition of an additional mutation, A77V, that doubled the infectivity of KAKA/T8I/G225R (Fig. 1i). Curiously, we also found that G225R conferred an opposing phenotype to KAKA, decreasing its infectivity to nearly undetectable levels (Fig. 1i). Thus, T8I, which stabilizes the immature Gag lattice, restores KAKA/G225R single-cycle infectivity by ~100 fold. Furthermore, the G225R mutation alone imposes an infectivity defect comparable to that of KAKA/T8I/G225R, demonstrating that G225R is insensitive to the defect imposed by KAKA/T8I (Fig. 1i).

## G225R does not restore IP6 enrichment to the KAKA or KAKA/T8I mutants

We next sought to determine the mechanism by which G225R restores infectivity to KAKA/T8I. Because G225 is close to the mutated IP6-binding residue K227 at the top of the 6HB, the change of the Gly to a positively charged Arg at this position could introduce a new positively charged ring capable of restoring IP6 recruitment during particle assembly, thereby rescuing particle infectivity. To investigate this possibility, we produced immature KAKA/G225R, KAKA/T8I/G225R, A77V/KAKA/T8I/G225R, and G225R VLPs and solved their Gag hexamer

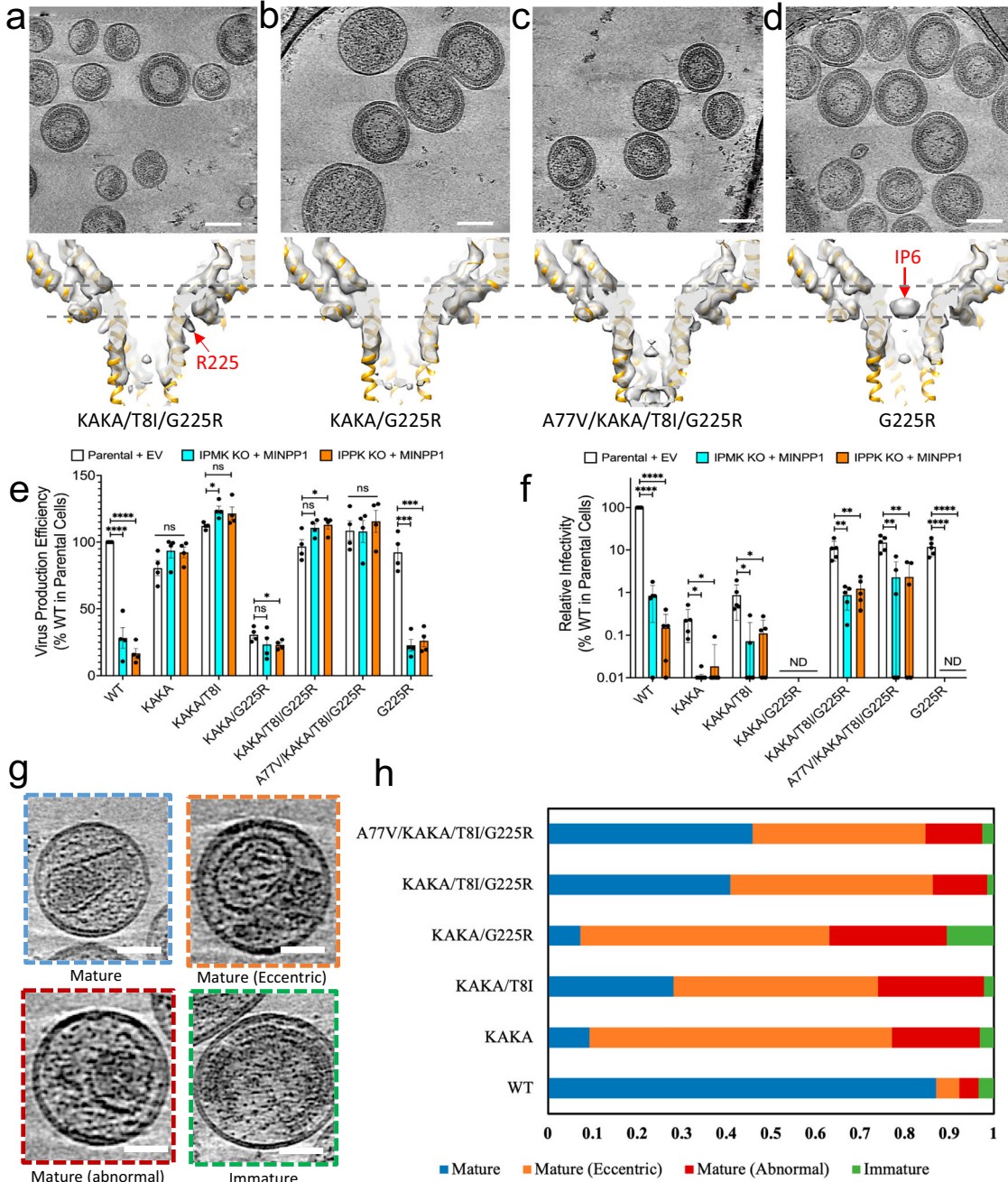

structures by cryoET with STA (Fig. 2a–d, Supplementary Figs. 1, 2, Supplementary Table 1). The structure of KAKA/T8I/G225R at 4.05 Å resolution revealed no IP6 density atop the 6HB (Fig. 2a, Supplementary Fig. 1c, Supplementary Table 1). The bulky Arg sidechain, which is clearly resolved in the density map, points away from the central channel of the 6HB, unlike K227 in WT Gag, and is therefore not available for IP6 coordination (Fig. 2a). Indeed, no IP6 density was observed in cryoET STA maps in any of the KAKA-containing Gag mutants analysed, including KAKA/G225R and A77V/KAKA/T8I/G225R (Fig. 2b, c, Supplementary Fig. 1d, e, Supplementary Table 1). In contrast, the G225R Gag hexamer structure showed a clear IP6 density atop the 6HB (Fig. 2d, Supplementary Fig. 1f, Supplementary Table 1). Each of the Gag mutants evaluated is capable of assembling an immature Gag lattice similar to that present in WT immature VLPs (Fig. 2a–d). These structural data suggest that the recovery of KAKA/T8I infectivity upon introduction of the compensatory mutation G225R cannot be attributed to increased IP6 packaging.

## G225R does not affect the production of KAKA/T8I virions from IP6-depleted cells

To confirm that the KAKA-containing mutants rescued by G225R remain IP6-independent during particle assembly, we conducted experiments to investigate the effect of IP6 depletion in virus-producing cells on virus production efficiency and particle infectivity. Briefly, we produced virus in HEK 293T knockout (KO) cell lines lacking either inositol polyphosphate multikinase (IPMK), which phosphorylates IP3 and IP4 to generate IP4 and IP5, or inositol pentakisphosphate 2-kinase (IPPK), which phosphorylates IP5 to generate IP6[13]. To further decrease IP6 levels in IPMK or IPPK KO cells, we simultaneously overexpressed multiple inositol-polyphosphate 1 (MINPP1), which removes phosphates from IP6, IP5, and IP4. After transfecting NL4-3 infectious molecular clones, we measured the efficiency of virus production in these IP6-depleted cells by quantifying the levels of p24 (CA) present in the supernatant relative to total Gag present in the cells and supernatant. Consistent with previous

**Fig. 2 | G225R does not restore IP6 recruitment in IP6 deficient mutants.**
**a**–**d** Representititve tomographic slices of KAKA/T8I/G225R, KAKA/G225R, A77V/
KAKA/T8I/G225R and G225R VLPs with slice thickness 4.02 nm (top) and their
corresponding cryoET STA structures of immature Gag hexamer (bottom). Scale
bar, 100 nm. **e** Cell lysates and concentrated virus lysates were collected after a 24 h
incubation following media change after co-transfection of pNL4−3 with 500 ng of
empty vector (EV) or MINPP1 expression vector (IPMK KO and IPPK KO). Cell and
virus Gag levels were quantified by western blot, and virus production efficiency
was calculated as described in the methods. Data are the mean of 4 independent
biological replicates for all groups ±SEM. Statistical analysis was performed using
GraphPad Prism. Statistical significance was determined by a two-tailed one-sample
Student's $t$ test with a hypothetical value set to 100 when comparing groups to
WT. All other comparisons were made using two-tailed unpaired Student's $t$ tests.
($p$-value summary: >0.05 = not significant; <0.05 = *; <0.01 = **; <0.001 = ***;
<0.0001 = ****. Precise $p$-values for each comparison−WT (Parental vs IPMK
$P < 0.0001$; Parental vs. IPPK $P < 0.0001$), KAKA (Parental vs. IPMK $P = 0.1473$; Par-
ental vs. IPPK $P = 0.1402$), KAKA/T8I (Parental vs. IPMK $P = 0.0107$; Parental vs. IPPK
$P = 0.1094$), KAKA/G225R (Parental vs. IPMK $P = 0.2634$; Parental vs. IPPK
$P = 0.0482$), KAKA/T8I/G225R (Parental vs. IPMK $P = 0.0594$; Parental vs. IPPK
$P = 0.0378$), A77V/KAKA/T8I/G225R (Parental vs. IPMK $P = 0.9613$; Parental vs. IPPK

$P = 0.5511$), G225R (Parental vs. IPMK $P = 0.0001$; Parental vs. IPPK $P = 0.0002$).
**f** Virus from cells transfected as above was collected 24 h post-transfection and
assessed for specific infectivity on TZM-bl cells. Infectivity was normalized to the
infectivity of WT produced from parental cells. Data are the mean of 4 independent
biological replicates for KAKA/G225R and 5 independent biological replicates for all
other groups. Replicates that did not produce a measurable signal were assigned a
value of 0 and are plotted at the base on the y-axis for visualization. Statistical
analysis was performed as in (**e**). Precise $p$-values for each comparison−WT
(Parental vs. IPMK $P < 0.0001$; Parental vs. IPPK $P < 0.0001$), KAKA (Parental vs.
IPMK $P = 0.0153$; Parental vs. IPPK $P = 0.0234$), KAKA/T8I (Parental vs. IPMK
$P = 0.0266$; Parental vs. IPPK $P = 0.0322$), KAKA/T8I/G225R (Parental vs. IPMK
$P = 0.0031$; Parental vs. IPPK $P = 0.0040$), A77V/KAKA/T8I/G225R (Parental vs.
IPMK $P = 0.0025$; Parental vs. IPPK $P = 0.0021$), G225R (Parental vs. IPMK $P = 0.0007$;
Parental vs. IPPK $P = 0.0007$). **g** A Gallery of distinct morphologies of HIV-1 particles
produced from HEK-293 cells, shown in tomographic slices. Particle morphologies
are classified as indicated. The slice thickness is 4.36 nm. Scale bar, 50 nm.
**h** Distribution of particle morphologies of WT ($n = 442$) and KAKA ($n = 162$), KAKA/
T8I ($n = 181$), KAKA/G225R ($n = 152$), KAKA/T8I/G225R ($n = 66$), A77V/KAKA/T8I/
G225R ($n = 687$) mutant Gag virions from one independent virus production.
Source data are provided as a Source Data file.

reports[20,21,31], we observed a significant reduction in the efficiency of
WT virus production from IPMK KO and IPPK KO HEK 293T
cells overexpressing MINPP1 relative to parental 293T cells co-
transfected with an empty vector (Fig. 2e, Supplementary Fig. 4). In
agreement with our structural data, G225R exhibited an IP6-dependent
assembly phenotype similar to WT. In contrast, virus production effi-
ciency of all KAKA-containing mutants was unaffected by producer-cell
IP6 depletion, confirming that G225R does not rescue KAKA/T8I
infectivity by restoring IP6 enrichment. Again, these data are in
agreement with our cryo-ET/STA results and demonstrate that irre-
spective of the presence of G225R, KAKA mutants assemble in an IP6-
independent manner.

We then investigated the effects of producer-cell IP6 depletion on
particle infectivity. Consistent with previous results[20,21,31] producer-cell
IP6 depletion drastically reduces WT particle infectivity, while KAKA
and KAKA/T8I infectivity is more mildly reduced (Fig. 2f). Our assay
was unable to detect KAKA/G225R infectivity under any condition. The
rescued mutants KAKA/T8I/G225R and A77V/KAKA/T8I/G225R also
showed only mild reductions in infectivity when produced from IP6-
depleted cells. However, KAKA/T8I/G225R virions produced from IP6-
depleted cells remained ~10−fold more infectious than KAKA/T8I vir-
ions produced under the same conditions (Fig. 2f). Since neither
mutant is able to enrich IP6 into virions, this suggests that G225R may
rescue KAKA/T8I infectivity by mitigating the consequences of
reduced IP6 during particle maturation, perhaps by facilitating capsid
formation at low IP6 concentrations.

## G225R increases the efficiency of KAKA/T8I capsid formation in virions and in vitro

To test the hypothesis that G225R facilitates capsid formation in the
absence of enriched IP6, we next examined the effects of G225R on the
morphology of mature, Env(-) viral particles by cryoET (Fig. 2g, h).
CryoET reconstructions revealed a variety of distinct morphologies of
mature WT HIV-1 particles produced from HEK 293T cells, which we
categorized into Mature, Mature (eccentric), Mature (abnormal), and
Immature (Fig. 2g). All the mutants tested displayed reduced numbers
of viral particles containing normal mature conical cores relative to the
WT. The revertant mutants KAKA/T8I/G225R and A77V/KAKA/T8I/
G225R exhibited higher numbers of viral particles containing normal
mature conical cores than KAKA/T8I (Fig. 2h). The occurrence of
normal mature cores (Fig. 2h, light blue) correlates well with the
observed infectivity measurements for these mutants (Fig. 1i). These
analyses demonstrate that G225R increases the formation of normal
mature cores in the absence of IP6 enrichment into viral particles.

The findings that the G225R mutation improves mature particle
formation without IP6 enrichment led us to hypothesize that this
mutation may increase the efficiency of mature capsid assembly under
low-IP6 conditions. To test this, we expressed and purified recombi-
nant CA protein carrying KAKA and G225R mutations and conducted
in vitro assembly assays at IP6 concentrations ranging from 0 μM to
5 mM (Fig. 3a−d). Assembly properties of each mutant were evaluated
by determining the amount of pelletable CA in each reaction. While WT
and KAKA CA require high IP6 concentrations to assemble (Fig. 3a, b),
the CA proteins carrying the G225R mutation assemble at low IP6
concentrations (Fig. 3c−e). Negative stain transmission EM images
show that the KAKA/G225R CA starts forming tubes and cones at an IP6
concentration as low as 5 μM and the G225R CA at 50 μM, whereas no
such structures were observed with WT and KAKA CA at 150 μM IP6
(Fig. 3f). Further statistical analyses on the number of assembled tubes
and cones for each variant indicate that KAKA/G225R and G225R CA
form significantly more tubes and cones than WT and KAKA CA at low
IP6 concentrations (Fig. 3g−i). The data suggest that the G225R
mutation reduces IP6 dependency for mature capsid assembly.

## CryoEM structure of KAKA/G225R CLP reveals previously unre-solved CA C-terminus stabilizing the dimer interface

To understand the molecular mechanism by which the G225R muta-
tion promotes the assembly of stable mature capsids under low-IP6
conditions, we prepared KAKA/G225R CLPs in the presence of 100 μM
IP6 to optimize the sample for cryoEM structural analysis, noting that
WT CA shows no assembly at this IP6 concentration (Fig. 3f−i). The in
vitro assembled KAKA/G225R CLPs are highly ordered (Fig. 4a), from
which we determined the structure of KAKA/G225R CA hexamer at
2.7 Å resolution and built an atomic model (PDB 9I8I) using single-
particle cryoEM (Fig. 4b, c, Supplementary Fig. 5, Supplementary
Table 3).

The structure reveals two IP6 densities in the hexamer center,
coordinated by R18 at the top density and K25 at the bottom density
(Fig. 4d), consistent with previous studies[4,6,7,9,32,33]. It is worth noting
that the IP6 densities are rather weak compared to those in WT CA
CLPs assembled with 2.5−5 mM IP6[9,32,33] and in native mature cores
within perforated virions with 1 mM IP6[7], suggesting a low IP6 occu-
pancy in KAKA/G225R CLPs. This is expected, as our KAKA/G225R CLPs
were assembled in the presence of 25 to 50−fold lower IP6 con-
centrations than WT CLPs. In the hexamer structure, the β-hairpin
exhibits an open conformation similar to those observed in previous
cryoEM structures derived from CLPs as well as the crystal structure
from the non-pandemic strain HIV-1 (O) at pH 6, distinct from the

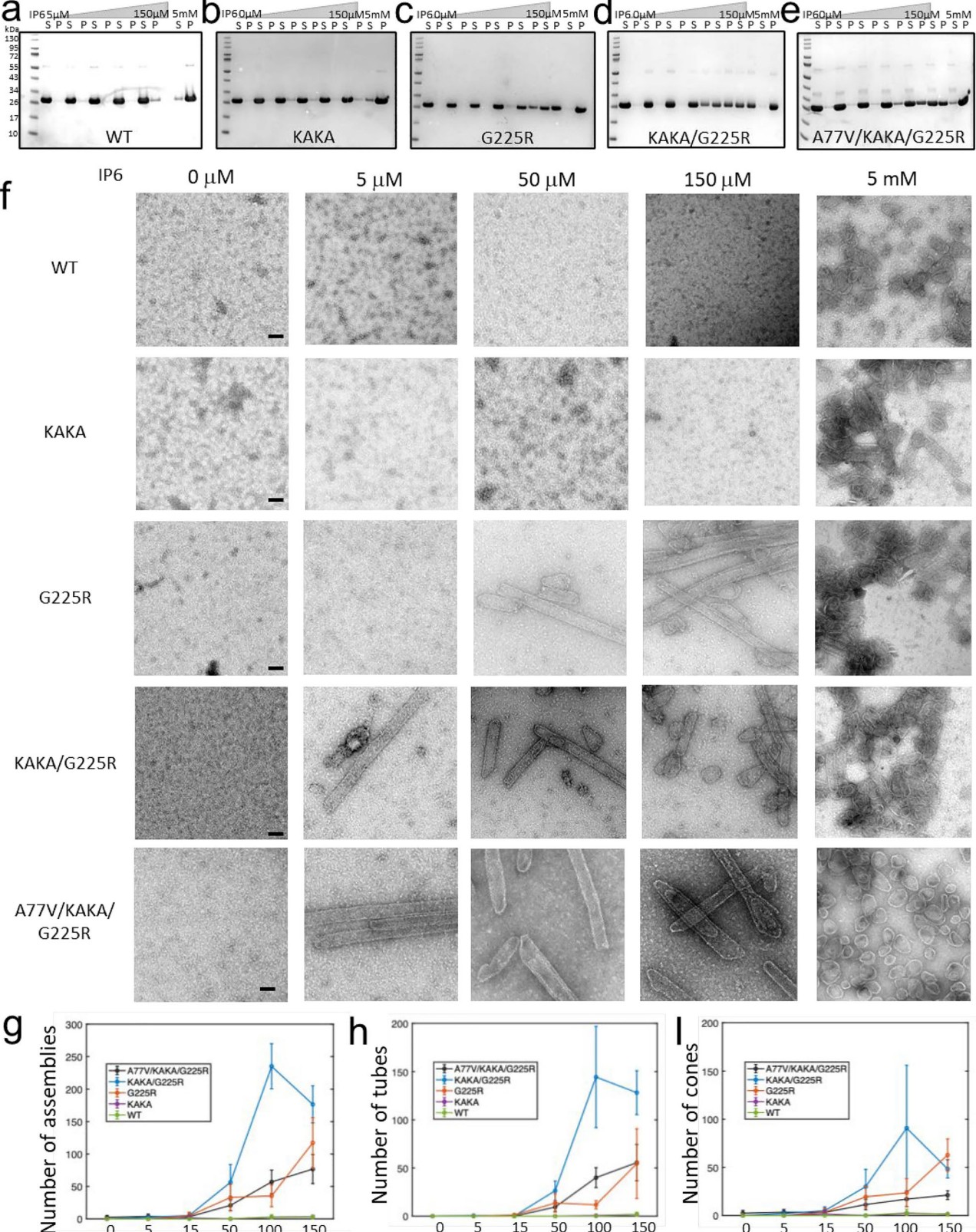

**Fig. 3 | G225R increases the efficiency of capsid formation in vitro. a–e** Sup-pellet assays for the in vitro assembly of CA WT, KAKA, G225R, KAKA/G225R, and A77V/KAKA/G225R at 0, 5, 15, 50, 150 μM and 5 mM IP6 concentrations (Three independent experiments were conducted with similar results for each mutant). **f** Negative stain images of CA WT, KAKA, G225R, KAKA/G225R and A77V/KAKA/G225R with IP6 concentrations from left to right are 0, 5, 50 and 150 μM, respectively (More than 10 micrographs were imaged for each condition). The numbers of total assemblies (**g**) and assembled tubes (**h**) and cones (**i**) in each micrograph of WT and mutant CA at different IP6 concentrations (Data are the mean of 5 independent micrographs for each condition and error bars depict ±STD). Scale bars, 100 nm. Source data are provided as a Source Data file.

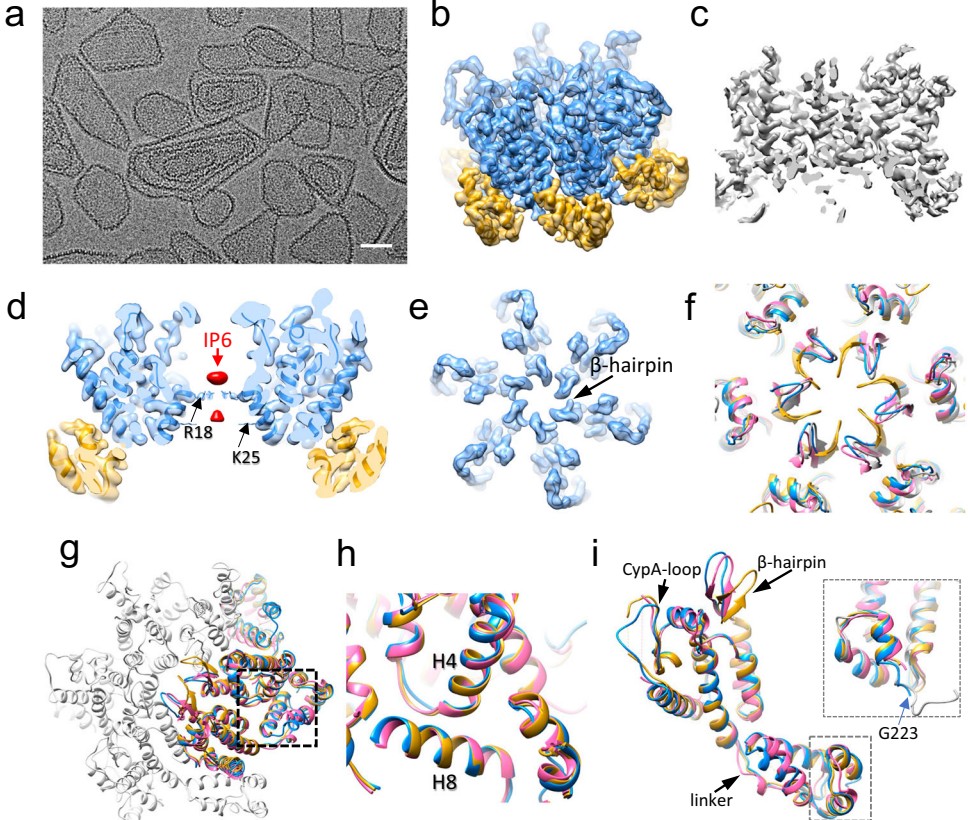

**Fig. 4 | CryoEM structure of CA KAKA/G225R hexamer. a** A representative cryoEM micrograph of in vitro assembled KAKA/G225R capsid with 100 μM IP6 (8443 micrographs were collected). **b** The overview KAKA/CA G225R density map at 2.7 Å resolution superimposed with the refined molecular model (PDB 9I8I), with CA NTD colored blue and CA CTD colored gold. **c** The central slice of KAKA/G225R CA density map. **d** The central slice of KAKA/G225R CA hexamer density map superimposed with the refined model with IP6 density colored pink. R18 sidechain is indicated. **e** The top view of KAKA/G225R CA hexamer density map super-imposed with the refined model, showing β-hairpins in an open conformation. **f** Comparison of β-hairpin among HIV-1 CA hexamers from crystal of WT (M) (PDB 5HGN, gold) and HIV-1 (O) (PDB 7T12, pink), cryoEM of WT CLP (PDB 7URN, grey) and KAKA/G225R (PDB 9I8I, blue). **g** Comparison of CA NTD-CTD interfaces among HIV-1 KAKA/G225R, HIV-1 WT (M) and HIV-1 (O). **h** An enlarged view of NTD-CTD interfaces. Helices H4 from CA1 NTD and H8 from CA2 CTD are labeled. **i** Comparison of CA monomers among HIV-1 KAKA/G225R (PDB 9I8I, blue), HIV-1 (M) WT (PDB 5HGN, gold) and HIV-1 (O) (PDB 7T12, pink). Inset: an enlarged view of the C-terminus, additionally overlapped with an NMR HIV-1 WT CTD$_{144-231}$ model (PDB 2KOD, grey). The last resolved residue G223 in the cryoEM structure (blue) is marked. Scale bar, 50 nm.

closed conformation in cryoEM structures determined from tubular assemblies and the crystal structure of the WT CA hexamer at pH 7[8,9,32–35] (Fig. 4e, f). The reduced pH in the CLP assemblies was suggested to affect the β-hairpin conformation[4,6,8–12]. The NTD-CTD interface around the binding pocket for Lenacapavir and FG-motif-containing host factors appears preserved (Fig. 4g, h). The linker that connects CA-NTD to CA-CTD resembles that of the WT CA (Fig. 4i). The CypA binding loop is very well ordered (Fig. 4i), in contrast to the majority of CA hexamer structures determined previously. Intriguingly, the density of KAKA/G225R CA C-terminus extends to G223 which can be unambiguously modeled (Fig. 4i, inset), along with additional densities further extending to the inter-hexamer interfaces but not well-resolved. Interestingly, this C-terminal extension adopts the same configuration as our previous CA-CTD NMR solution structure (PDB 2KOD, the 2nd conformer) (Fig. 4i inset, grey)[36]. These results suggest that the otherwise flexible C-terminus could adopt a structured conformation to mediate intermolecular interactions, thus strengthening the capsid stability when required.

To further clarify the inter-hexamer densities, cryoEM density maps of the tri-hexamer interface were obtained (Fig. 5a). The dimer and trimer interfaces of KAKA/G225R are very similar to those of WT (PDB 8G6M), with an RMSD of 0.27 Å and 0.44 Å, respectively (Fig. 5b, c). A further 3D classification and refinement resulted in two major classes, one without extra density (Fig. 5d) and another with an

extra density extending from the end of H11, reaching below the dimer interface (Fig. 5e, red density). This density most likely corresponds to the previously unresolved CA C-terminal fragment (220–231), which was too flexible to be resolved in all previous crystal or cryoEM structures.

## Molecular dynamics simulation of interactions involving the CA C-terminal segment

To explore the interactions between the C-terminal fragment with the neighboring CA monomers, we used Rosetta[37–39] to derive density-guided models of a KAKA/G225R CA trimer of dimers with an extended C-terminus. Furthermore, we performed molecular dynamics simulations of the KAKA/G225R CA trimer of dimers, as well as a KAKA CA and WT CA trimer of dimers, to assess the stability of the C-terminal fragment in the CA dimer interface. Across 7 independent replicates of 400 ns long unrestrained equilibration MD simulations, we observed that the C-terminal region (residues 220–231) is highly flexible and transiently occupies the CA dimer interface. In the KAKA/G225R CA trimer of dimers simulations, C-terminal tail occupancies at the dimer interface ranged from 1 to 90%, with an average occupancy of 20.7%, consistent with the multiple 3D classes observed in the cryoEM analysis. In contrast, for KAKA CA and WT CA, the dimer interface occupancies were consistently lower, with average occupancies of 6.9% and 10.4%, respectively (Supplementary Fig. 6a, b).

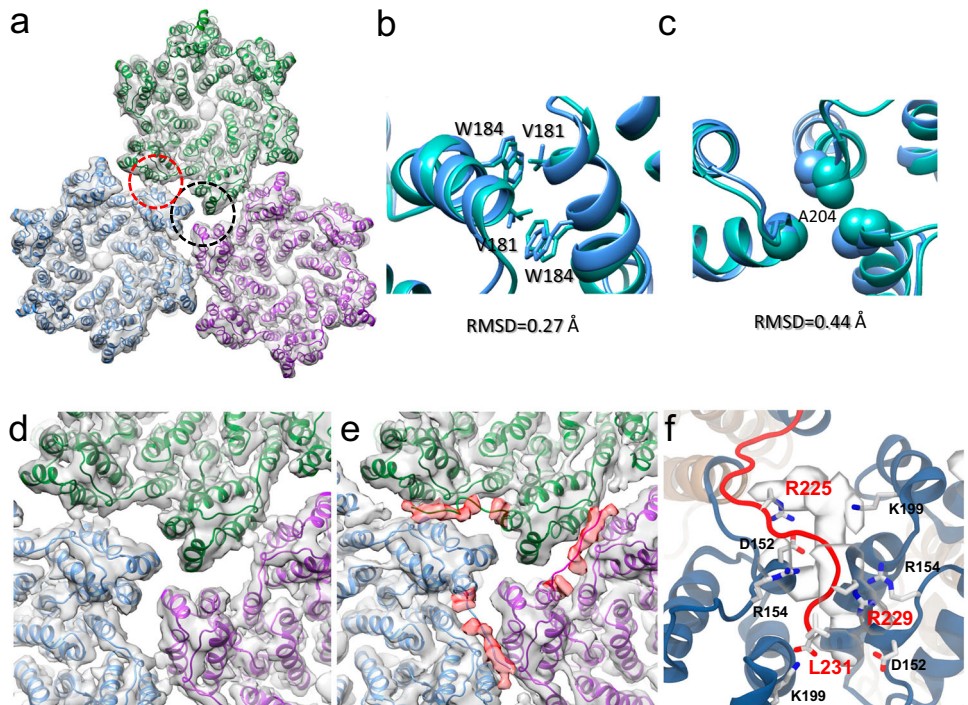

**Fig. 5 | CryoEM structures of CA KAKA/G225R tri-hexamer. a** The overview KAKA/CA G225R tri-hexamer density map at 3.63 Å resolution superimposed with the refined molecular model (PDB 9I8I). Three hexamer models are colored blue, green and purple. The dimer and trimer interfaces are marked with red and black dashed circles. **b** Comparison of dimer interfaces between WT (PDB 6SKN, green) and KAKA/G225R (PDB 9I8I, blue). **c** Comparison of trimer interfaces between WT (PDB 6SKN, green) and KAKA/G225R (PDB 9I8I, blue). Two 3D classes of CA KAKA/ G225R tri-hexamer maps, superimposed with model (PDB 9I8I), viewed from CTD side (inside of capsid). Class 2 **e** distinguishes from Class 1 **d** by additional densities (segmented in red) extending from H11 to the dimer interface, which is overlaid with an MDFF model of the C-terminus (220–231). **f** Interactions observed between C-terminal region (220–231, red) and residues at the dimer interface in unbiased molecular dynamics simulation of the KAKA/G225R CA trimer of dimers superimposed with the density map. Interacting residues are labeled.

Additionally, we analysed the residence times of C-terminal segment contacts in all simulations. While C-terminal interactions at the dimer interface generally have short residence times (<5 ns) across all CA variants, contact events for KAKA/G225R CA exhibited a broader distribution of residence times, with several events lasting over 20 ns and up to 150 ns (Supplementary Fig. 6c). Interestingly, across all simulation replicas we also observed one high residence time event for WT CA, where the C-terminal tail was stabilized via salt bridge interactions involving R229 and L231 with charged residues at the dimer interface. This suggests that, although C-terminal tail interactions with the dimer interface can occur in WT CA, these events are rare and more prevalent in KAKA/G225R CA.

We then performed a residue-level contact analysis of the C-terminal segment and CA dimer interactions to identify the specific interactions that stabilize the C-terminal tail at the dimer interface. Notably, we observed that residues L231 (the C-terminal residue), R229 and G225R form salt bridges with high occupancies with residues K199, R154 and D152 at the dimer interface (Fig. 5f, Supplementary Table 4). These interactions increase the stability of conformations where the C-terminal segment resides at the dimer interface. In contrast, interactions with those residues have significantly reduced occupancies in the KAKA CA and WT CA simulations (Supplementary Table 4).

Thus, the extra density observed in the cryoEM map of KAKA/ G225R CA tri-hexamer likely stems from the transient interactions of the C-terminal segment at the dimer interface, which are facilitated by the added charge of the G225R mutation on the C-terminal segment. The presence of the C-terminal segment at the dimer interface, along with the formation of salt bridge interactions therein, might explain why the KAKA/G225R CA mutation stabilizes the mature capsid even with minimal IP6 present.

## KAKA/G225R CA forms compact assembly interfaces

To further evaluate the effect of the KAKA/G225R CA mutation in capsid interfaces, we extracted all possible capsomer-capsomer interfaces (hexamer-hexamer-hexamer, hexamer-hexamer-pentamer and hexamer-pentamer-pentamer) from a full capsid cone[7] and compared them with the KAKA/G225R CA trimer of dimers assembly (Fig. 6). For each capsomer-capsomer dimer interface, we measured the center of mass distances and tilt angles between alpha helices 9 and 10 and the 3–10 helix. Compared to all WT dimer interfaces, the helices 9 and 10 in KAKA/G225R CA are packed closer, resembling the distance for pentamer-pentamer-hexamer interfaces. However, these helices are oriented at a more acute angle (for helix 9), or closer to a right angle (for helix 10). In conclusion, the KAKA/G225R CA dimer interfaces are more tightly packed than those measured in a WT CA cone, likely due to the interactions with the C-terminal flexible tail described above.

## Infectivity of the KAKA and G225R mutants is insensitive to target-cell IP6 depletion

We further investigated whether an inability to enrich IP6 into viral particles results in sensitivity to depletion of IP6 in target cells, and whether G225R could reverse this sensitivity. Consistent with previous reports[40], we found that the infectivity of the hypostable CA mutant P38A was severely reduced in HEK 293T IPPK KO target cells relative to its infectivity in parental HEK 293T cells (Supplementary Fig. 7). In contrast, WT and KAKA/T8I infectivity was unaffected in HEK 293T IPPK KO cells relative to parental cells, and the addition of G225R to KAKA/T8I did not alter this phenotype. We did observe a statistically significant decrease in G225R infectivity in HEK 293T IPPK KO cells relative to parental cells. However, the effect size was much smaller

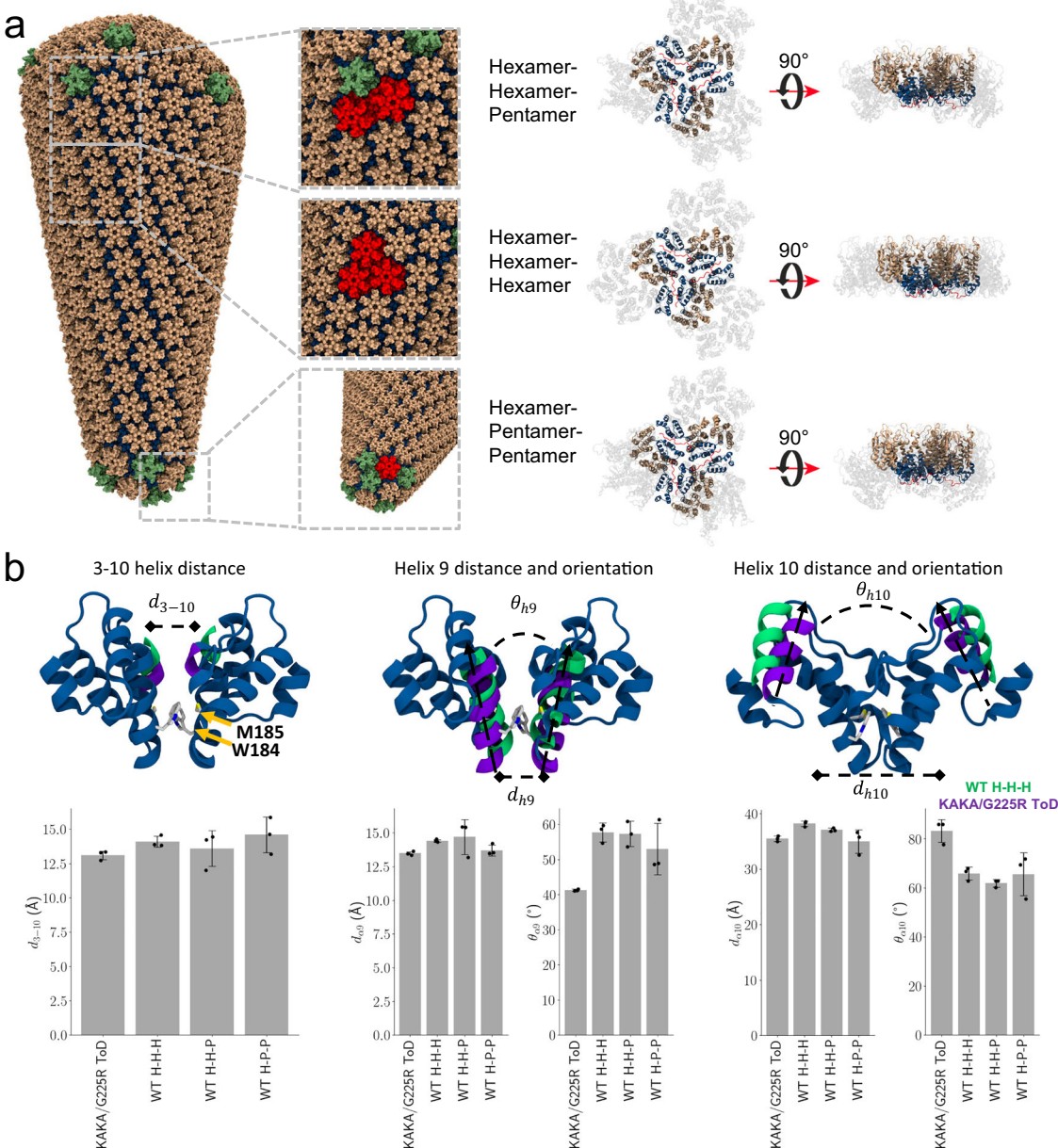

**Fig. 6 | KAKA/G225R CA dimer interfaces compared to assembly interfaces from full size capsid cone. a** All cases of capsomer interfaces in the full WT capsid (left) overlayed on the KAKA/G225R CA trimer of dimers (right). NTD is colored tan on hexamers and green on pentamers; CTD is colored dark blue. **b** Distance and orientation measurements of the CTD structural elements at the dimer interface for the KAKA/G225R CA trimer of dimers (ToD) or the WT capsomer interfaces involving hexamers (H) and pentamers (P). Bar plots represent mean values over all dimers in a trimer of dimers ($n = 3$), and error bars represent standard deviation. Helices for a WT hexamer-hexamer-hexamer are colored in teal, and helices for the KAKA/G225R CA ToD are colored in purple. Dimer interface residues M185 and W184 are visualized in licorice representation. Source data are provided as a Source Data file.

than that of P38A. These data demonstrate that the addition of G225R to KAKA/T8I did not eliminate insensitivity to target cell IP6 depletion.

## Discussion

Here, we investigated how HIV-1 adapts to an inability to specifically package and enrich IP6 into particles during virus assembly. Propagation of the IP6-binding-deficient mutants KAKA and KAKA/T8I resulted in the acquisition of a mutation in the C-terminus of CA−G225R− that rescued KAKA/T8I particle infectivity by increasing the efficiency of capsid formation in virions. In vitro assembly assays revealed that CA protein harboring the G225R mutation assembled into CA tubes and cones at substantially lower IP6 concentrations than required for WT assembly. These findings suggest that G225R

facilitates capsid assembly in virions containing limited IP6, explaining the ability of G225R to restore KAKA/T8I capsid formation and infectivity. Importantly, there was no indication of CA assembly in the absence of IP6, suggesting that G225R does not confer fully IP6-independent assembly to CA. Some IP6 is still required to promote mature capsid assembly, especially the conical-shaped capsids (Fig. 3). In a biological context, virions unable to actively incorporate IP6 during assembly, such as KAKA/T8I, likely have access to low levels of passively incorporated IP6. Based on our structural, morphological, and in vitro assembly data, we propose that G225R restores KAKA/T8I capsid assembly and infectivity by drawing upon the small pool of passively incorporated IP6 available during maturation, considering that the cellular IP6 concentration is around 50−100 μM, which is

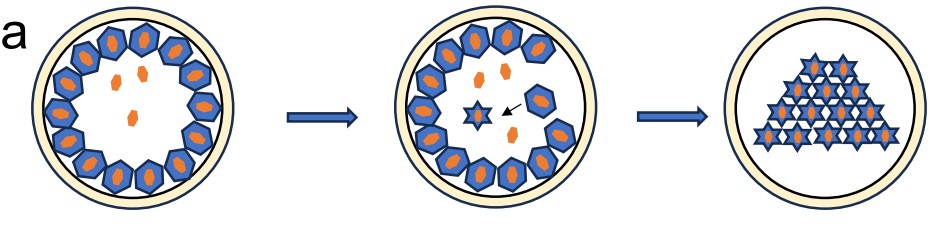

IP6 promotes immature lattice to mature capsid cone

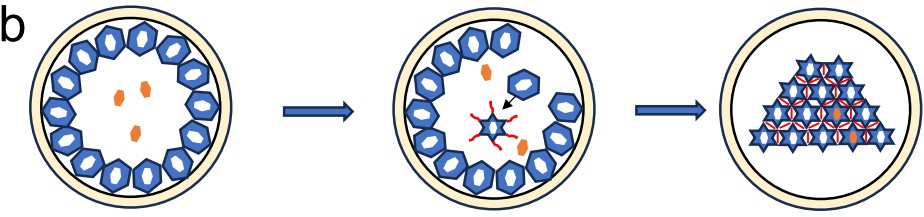

CA unstructured C-terminus in KAKA/G225R promotes mature capsid cone assembly

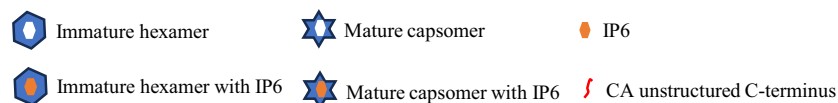

**Fig. 7 | A schematic model for the role of G225R in HIV-1 mature capsid assembly. a** The assembly WT mature capsid with high level of IP6 enrichment through the immature Gag assembly. **b** The assembly KAKA/G225R mature capsid with low level of cytosolic IP6, as the mutant is defective in IP6 packaging by immature Gag. The assembly of mutant mature capsid is facilitated by the interactions with the CA C-terminal fragment which is otherwise unstructured and flexible.

sufficient to assemble KAKA/G225R capsids. To address the mechanism by which G225R increases the efficiency of capsid assembly, we solved a tri-hexamer structure of in vitro assembled KAKA/G225R CLPs. We observed a strip of electron density corresponding to the CA C-terminus extending from the end of helix 11 at the trimer interface between hexamers into the dimer interface between hexamers. Molecular dynamics analysis revealed that the KAKA/G225R mutations promote the formation of transient salt bridges between residues in the C-terminal segment (residues 220–231) and residues in helix 10 and the 3–10 helix at the dimer interface, in particular G225R and R229 with D152. These interactions increase the occupancy of the C-terminus at the dimer interface and contribute to forming a more compact interface. These results demonstrate that the unstructured and flexible CA C-terminus is capable of altering the properties of mature CA assembly and may indicate a role for the CA C-terminus during HIV-1 particle maturation (Fig. 7).

The stability of the HIV-1 capsid is critical for several post-entry functions, including reverse transcription, nuclear entry, and interactions with host factors. Thus, the assembly of a stable capsid during maturation is required to set the stage for these key steps in the HIV-1 replication cycle. IP6, which is packaged into virions via its interaction with the Gag hexamers that compose the immature Gag lattice, plays a significant role in mature capsid assembly and the maintenance of capsid stability. Our observation of a mutation in the CA C-terminus that increases the efficiency of capsid formation in IP6-packaging-deficient virions and CLP assembly in vitro at low IP6 concentrations suggests that the flexible C-terminus may allosterically regulate capsid assembly and stability.

IP6 has also been proposed to play a critical role in CA pentamer formation, a requirement for the assembly of a closed conical capsid[7,9–11,18,32,33,41]. It is possible that mutations in the C-terminus of CA may promote pentamer formation under limiting IP6 conditions. The mutation T216I, which lies in helix 11 near the unstructured C-terminus, was recently proposed to reverse a defect in CA pentamer formation induced by mutation of the IP6-interacting residue K25[11] and was also described as a compensatory mutation for the hypostable capsid mutant P38A[42]. Interestingly, T216I also arose during propagation of the IP6-packaging-deficient mutants KAKA and KAKA/T8I in this study (Supplementary Table 2), though its addition to KAKA/T8I only mildly restored infectivity (Supplementary Fig. 3). Propagation of IP6-packaging-deficient mutants also resulted in the acquisition of mutations at several other amino acid positions previously implicated in pentamer formation. These include the N21S mutation within the central pore, which exhibits the same phenotype described above for T216I[11]; the T58I and G61E mutations, which are part of the so-called "Thr-Val-Gly-Gly (TVGG) switch" that controls pentamer formation[33]; and the M68I mutation, which lies near the Met-66 residue that plays a critical role in the conformation of the TVGG switch[33].

While the data clearly show that G225R enhances mature capsid assembly in vitro and in virions, the size of the effect in virions does not fully explain the observed increase in infectivity. Upon the addition of G225R, we observed a roughly 30% increase in the number of KAKA/T8I virions containing mature conical capsids. However, KAKA/T8I/G225R is ~8–fold more infectious than KAKA/T8I. This suggests that KAKA/T8I/G225R capsids are inherently more stable than KAKA/T8I capsids. We also note that propagation of KAKA/T8I/G225R resulted in the acquisition of the A77V mutation, which doubled the infectivity of KAKA/T8I/G225R to ~30% of WT infectivity. A77 lies at the NTD-CTD interface near a binding pocket critical for HIV-1 capsid interactions with host factors, including CPSF6[43], Nup153[44], and Sec24C[45], and capsid inhibitors such as PF74[46] and lenacapavir (LEN)[47]. Interestingly, A77V has been previously described as a CPSF6-binding-deficient mutant that maintains WT-level fitness in primary cells and humanized mice[48]. The interaction between the HIV-1 capsid and CPSF6 has been implicated in several viral functions, including nuclear entry, uncoating, and integration site selection[49–51]. CPSF6-binding-deficient mutant cores have been shown to dock at nuclear pores but not enter the nucleus, completing reverse transcription and uncoating at the

nuclear envelope[50]. It is unclear how a loss of CPSF6 binding would increase the infectivity of KAKA/T8I/G225R. One possibility is that KAKA/T8I/G225R cores are not stable enough to efficiently pass through nuclear pores intact, and the addition of A77V spares these cores the stress of passing through nuclear pores, allowing reverse transcription and uncoating to occur more efficiently. Alternatively, A77V may simply enhance capsid assembly or the stability of KAKA/T8I/G225R cores or perform some other infectivity-promoting function in this context. The mutants found in these selections will provide fertile ground for further virological and structural investigation.

While the G225R mutation occurs very infrequently in a natural context (observed in 3 of 9494 major subtype sequences analyzed online via the AnalyzeAlign tool provided by Los Alamos National Laboratory (LANL) HIV Sequence Database), it should be noted that its acquisition occurred through an initial G225S change. G225S is a frequent polymorphism, occurring at a rate of 27.8% in subtype B and 50% of subtype C sequences. G225S has arisen as a compensatory mutation in several different contexts, including as a CTL-escape mutation[52], and a compensatory mutation restoring replication to the SP1-A3V mutant[53], which demonstrates resistance to the maturation inhibitor, PA-457 (later named bevirimat [BVM]). Maturation inhibitors block CA-SP1 processing by stabilizing the 6HB of Gag hexamers in the immature lattice, leading to defects in capsid assembly during maturation. G225S restores replication to A3V without reversing its defect in CA-SP1 processing. We also recently reported that G225S was acquired upon propagation of CA-K25A[11], a mutant that is unable to assemble stable capsids. While G225S only mildly restores K25A infectivity, these findings suggest that G225S confers a restorative effect on capsid assembly and stability in non-optimal conditions. G225D was also previously identified as an assembly-deficient mutant whose replication could be restored by the maturation inhibitor PF-46396[17]. This suggests that G225 is critical for proper immature Gag lattice and virus assembly. These previous findings are consistent with our results, which suggest a role for G225 in the assembly of both the immature Gag lattice and capsid.

We observed that G225R-mediated rescue of capsid assembly in low IP6 conditions was dependent on the presence of the SP1-T8I mutation. This is curious because T8I did not restore IP6 enrichment in the context of KAKA or KAKA/G225R (Figs. 1g, h and 2a, b) and does not participate in capsid assembly after its removal from CA during the final step of Gag processing. G225R drastically reduces KAKA virus production efficiency and infectivity, while having no effect on KAKA/T8I virus production efficiency and increasing its infectivity (Figs. 1k and 2e, f). Our interpretation of these results is that G225R imposes a destabilizing effect on IP6-independent immature Gag lattice assembly that is restored by the addition of the T8I mutation. T8I promotes efficient IP6-independent particle assembly, and G225R subsequently promotes efficient capsid formation at reduced IP6 concentrations during particle maturation. These results suggest that the G225R mutation imposes contrasting IP6-dependent effects on immature and mature assembly. Interestingly, the T216I mutation demonstrated the same T8I-dependent phenotype as G225R (Supplementary Fig. 3). Together, these findings highlight the evolutionary pressure faced by HIV-1 CA to balance immature and mature assembly to maximize virion production and infectivity.

## Methods

### Cells

MT-4 (ARP-120), SupT1 (ARP-100), and C8166 (ARP-404) human T cell lines were acquired from American Type Culture Collection (ATCC) and cultured at 37 °C with 5% $CO_2$ in Roswell Park Memorial Institute (RPMI) 1640 medium (Corning) supplemented with 10% fetal bovine serum (GenClone), 2 mM L-glutamine, penicillin (100 U/mL; Gibco), and streptomycin (100 µg/mL). HEK 293T (CRL-3216) cells were acquired from ATCC and cultured in Dulbecco's modified Eagle's

medium (DMEM) supplemented as above. HEK 293T Parental, IPMK KO, and IPPK KO cells, described in previous studies[13,20,21], were cultured in DMEM supplemented as above.

### Plasmids

pNL4-3, a lab adapted, subtype B infectious molecular clone, was used for transfection of MT-4, C8166, and SupT1 T-cell lines to initiate replication and forced evolution experiments. pNL4-3 was also used to transfect 293T cells to produce virus for virus production efficiency and infectivity experiments. pNL4-3 plasmids harboring deletions in the *pol* and *env* genes (pNL4-3 ΔpolΔenv) were used to transfect 293T cells for the production of immature virions for cryo-ET analysis. pNL4-3 KFS (env-) was used to transfect HEK 293T cells to produce mature virions for cryo-ET analysis. An env-/vpr-, luciferase-encoding pNL4-3 plasmid (pNL4-3.Luc.R-.E-) was co-transfected into 293T cells along with a plasmid encoding VSV-G to generate VSV-G pseudo-typed HIV-1 particles for infectivity experiments in 293T Parental and IPPK KO target cells. HIV-1 CA mutants were generated by subcloning of DNA fragments purchased from Twist Bioscience.

### Forced-evolution experiments

MT-4 or C8166 cells were transfected with WT or mutant replication-competent pNL4-3 infectious molecular clones. Replication kinetics were quantified using a 32P–based RT assay as described previously[54]. The virus collected at the peak of replication was used to infect fresh cells for further propagation. Genomic DNA isolated from cells collected at the peak of replication was used as the template for PCR amplification of the HIV-1 Gag-coding region. PCR-amplified Gag fragments were treated with ExoSAP-IT PCR product cleanup reagent (Thermo Fisher Scientific Catalog #78201.1 ML) and subjected to Sanger sequencing to identify potential compensatory mutations. Selected mutations were incorporated into pNL4-3 in the presence and absence of the original mutations by subcloning Gag fragments purchased from Twist Bioscience. The resultant molecular clones were used to transfect MT-4 cells to measure their inherent replication kinetics.

### Virus production efficiency

Parental, IPMK KO, and IPPK KO HEK 293T cells were co-transfected with pNL4-3 WT or mutants plasmids with an expression plasmid encoding MINPP1 (IPMK KO and IPPK KO) or and empty vector (Parental) as described previously[20]. Viruses were produced for 24 h at 37 °C with 5% $CO_2$. Viral supernatants were filtered through a 0.45 µm syringe filter, ultracentrifuged and virus pellets and cell monolayers were lysed and subjected to western blot analysis. Virion-associated and cell-associated Gag was detected using the HIV-Immunoglobulin (HIV-Ig; BEI Resources ARP-3957) diluted 1:10,000 in TBST as a primary antibody overnight. Blots were then probed with an anti-human IgG horseradish peroxidase (HRP)-tagged secondary antibody (Sigma-Aldrich GAENA933) diluted 1:10,000 and imaged using a Sapphire Biomolecular Imager (Azure Biosystems), and quantified using Azure Spot analysis software. Virus production efficiency was calculated using the formula below and normalized to WT virus production efficiency in parental cells.

$$\frac{virus\,p24}{virus\,p24 + cell\,p24 + cell\,p41 + cell\,Pr55}$$

### Single-cycle infectivity

Parental, IPMK KO, or IPPK KO HEK 293T cells were transfected as above. Virus present in filtered supernatants was quantified by RT assay, and single-cycle infectivity assays were performed as described previously[20]. Briefly, virus stocks were serially diluted and used to infect TZM-bl cells in the presence of 10 µg/mL DEAE-dextran. Specific

infectivity was determined by measuring luminescence after lysis of the infected cells 36–48 h post-infection. All infectivity measurements were expressed as a percentage of the infectivity of WT virions produced from parental HEK 293T cells. To measure single-cycle infectivity in IP6 depleted target cells, HEK 293T cells were co-transfected with pNL4-3.Luc.R-.E- and a plasmid expressing VSV-G. VSV-G-pseudo-typed, replication-incompetent virus was quantified by RT assay were used to infect HEK 293T Parental or IPPK KO target cells as described above.

## Purification of mature and immature virions
HEK 293T cells were transfected with the pNL4-3-derived vectors using GenJet (Ver. II) transfection reagent as per the manufacturers protocol. The media was changed 8–16 h post-transfection and virions were produced for 36–48 h at 37 °C with 5% $CO_2$. Supernatants were centrifuged briefly and filtered through a 0.45 µm syringe filter to remove cellular debris. Virions were pelleted by ultracentrifugation through an 8% iodixanol cushion and resuspended in phosphate-buffered saline (PBS). Virions were then purified by ultracentrifugation through a 10–30% iodixanol gradient, diluted with PBS, and subjected to additional centrifugation step. Pelleted virions were resuspended in a small volume of PBS and loaded onto grids for cryo-ET analysis.

## CA protein purification and assembly
The cDNAs encoding HIV-1 capsid (CA) with G225R, K158A/K227A and K158A/G225R/K227A (KAKA/G225R) were mutated from the CA-pNL4-3-pET21 vector and verified by sequencing. Mutated CA proteins were purified as described in Liu et al.[55] and concentrated to 30 mg/ml. For CA cones/tubes assembly, 5 µl mutated CA proteins, 5 µl 500 mM MES pH = 6.0 and 5 µl IP6 at a desired concentration were mixed and incubated at 37 °C for 1 h. For supernatant and pellet assay, 10 µl of each assembled mixture was centrifuged at 12,000 × $g$ for 10 min, supernatants and pellets were separated and diluted 1:10 with SDS loading dye before loading to SDS PAGE.

## Negative stain grids preparation and image
For negative stain grids preparation, 400 mesh carbon coated copper grids were glow discarded. 3 µl assembled CA mutation mixture was loaded onto the grid, incubated for 1 min. The grid was then blotted to dry and incubated with uranyl formate solution for 1 min. The grid was blotted to dry again and loaded to a JEOL 120 keV TEM (Sir William Dunn School of Pathology, University of Oxford) for imaging. Number of assembled tubes and cones was counted from 5 micrographs recorded from each sample and averaged.

## CryoET grids preparation
To prepare frozen hydrated grids for tomography analysis using cryo-electron microscopy analysis, vitrification was performed using a Vitrobot (FEI, Thermo Scientifics) operating at 4 °C and 98% relative humidity (RH). 10 µl of concentrated gold fiducial was prepared by removing 90 µl of supernatant after the centrifugation of 100 µl of gold fiducial (Electron Microscopy Sciences, Inc.) at 25,000 × $g$ for 30 min at 4 °C. 3 µl of freshly prepared HIV virions and 6 nm size of gold nanoparticles (Electron Microscopy Sciences, Inc.) was mixed at a ratio of 9:1 (v/v) was applied to lacey carbon film copper 300 mesh, 100 micron grids (Electron Microscopy Sciences, Inc.) which was previously glow discharged at 15 mA for 5 s with a Pelco easiGlow Glow discharge cleaning system (Ted Pella, Inc.). After 5 s, grids were blotted for 2 s and plunged into liquid ethane. Frozen sample grids were preserved under liquid nitrogen temperature for imaging.

## CryoET data collection and subtomogram averaging
For K227A and KAKA/T8I samples, cryo-ET data collection was acquired using a Thermo Fisher Titan Krios microscope operated at 300 keV equipped with a Gatan Quantum post-column energy filter (Gatan Inc) operated in zero-loss mode with 20 eV slit width, and Gatan K3 direct electron detector in eBIC (Electron BioImaging Center, Diamond). Tilt series images were collected with SerialEM[56] with a nominal magnification of 64k and a physical pixel size of 1.34 Å per pixel. For G225R, KAKA/G225R, KAKA/T8I/G225R and A77V/KAKA/T8I/G225R samples, cryo-ET data collection was acquired using a Thermo Fisher Titan Krios microscope operated at 300 keV equipped with Facon4i detector and Selectrix with 10 eV slit width in eBIC (Electron BioImaging Center, Diamond). All the datasets were acquired using a dose-symmetric tilt-scheme[57] starting from 0° with a 3° tilt increment by a group of 3 and an angular range of ±60°. The accumulated dose of each tilt series was around 123 e⁻/Å² with a defocus range between −1.5 and −6 µm. In total, 71 tilt series from KAKA sample, 81 tilt series from KAKA/T8I sample, 78 tilt series from G225R sample, 86 tilt series from KAKA/G225R sample, 61 tilt series from KAKA/T8I/G225R and 89 tilt series from A77V/KAKA/T8I/G225R sample were collected. Each projection image was dose-fractioned into ten frames. Details of data collection parameters are listed in Supplementary Table 1.

The automated cryo-ET pipeline developed in-house was used for pre-processing (https://github.com/ffyr2w/cet_toolbox) through performing motion correction[58] of the raw frames, tilt-series alignment with IMOD[59]. The gold beads were manually inspected to ensure the centering of fiducial markers for each tilt series in eTOMO.

Subtomogram averaging for all the datasets was performed following the workflow of emClarity[60,61]. The in vitro assembled HIV-1 Gag structure (EMD-8403)[62] was low-pass filtered to 30 Å and used as the initial template for template search in 6x binned tomograms with a pixel size of 8.04 Å for K227A and KAKA/T8I samples, 9.006 Å for G225R, KAKA/G225R, KAKA/T8I/G225R and A77V/KAKA/T8I/G225R samples. 141,486 subtomograms based on the initial position and orientations were selected from 71 tilt series for K227A dataset, 159,659 subtomograms were selected from 81 tilt series for the KAKA/T8I dataset, 284,966 subtomograms were selected from 78 tilt series for the G225R dataset, 165,561 subtomograms were selected from 86 tilt series for the KAKA/G225R dataset, 116,132 subtomograms were selected from 61 tilt series for the KAKA/T8I/G225R dataset, and 228,256 subtomograms were selected from 89 tilt series for the A77V/KAKA/T8I/G225R dataset. The subtomogram averaging and alignment were performed iteratively using 6×, 5×, 4×, 3×, 2×, and 1× binned tomograms. A cylindrical alignment mask including 7 hexamers and a 6-fold symmetry was applied throughout the whole procedure. The final density maps were reconstructed at bin 1 and sharpened with a b-factor of −50.

## CryoET data collection and analysis of mature VLPs
Cryo-ET data collection was acquired using a Thermo Fisher Titan Krios microscope operated at 300 keV equipped with a Gatan Quantum post-column energy filter (Gatan Inc) operated in zero-loss mode with 20 eV slit width, and Gatan K3 direct electron detector in eBIC (Electron BioImaging Center, Diamond). Tilt series images were collected with SerialEM[56] with a nominal magnification of 42k and a physical pixel size of 2.18 Å per pixel. All the datasets were acquired using a dose-symmetric tilt-scheme[57] starting from 0° with a 3° tilt increment by a group of 3 and an angular range of ±60°. The accumulated dose of each tilt series was around 123 e⁻/Å² with a defocus range between −1.5 and −6 µm. In total, ~30 tilt series from each variant were collected. Each projection image was dose-fractioned into ten frames.

The automated cryo-ET pipeline developed in-house was used for pre-processing (https://github.com/ffyr2w/cet_toolbox) through performing motion correction[58] of the raw frames, tilt-series alignment with IMOD[59]. The gold beads were manually inspected to ensure the centering of fiducial markers for each tilt series in eTOMO. The morphological analysis was conducted manually from reconstructed 3D tomograms.

## CryoEM data collection and processing

CryoEM movie data were collected using EPU software on a Gatan K3 direct detector camera in super-resolution mode. Each movie contains 40 frames with an accumulated dose of 40 electrons/Å². The calibrated physical pixel size and the super-resolution pixel size were 1.34 and 0.67 Å per pixel, respectively. The defocus was prescribed in the range from −0.8 to −2.5 μm. A total of 8443 movies in super-resolution mode were collected for data analysis.

All frames of the raw movies were corrected for their gain using a gain reference recorded within 3 mmos of the acquired movie to generate a single micrograph using CryoSPARC Patch Motion Correction. The micrographs were used for the determination of the actual defocus using CryoSPARC Patch CTF Estimation. 14,047,760 particles were picked and extracted using CryoSPARC Template Picker, in which the templates were averaged from manually picked ~1000 particles. 4,195,547 particles were selected after several rounds of CryoSPARC 2D classification. CryoSPARC heterogeneous refinement was conducted with C6 symmetry applied. One class with 1,646,193 particles was selected for non-uniform refinement. The final hexamer structure was obtained by 3D reconstruction at super-resolution mode to a global resolution of 2.75 Å.

Then the particles were re-centered on the trimer interface by moving the coordinates for further heterogeneous refinement. One class of 1,320,752 particles without CA C-terminus density was selected for non-uniform refinement to the final map with a global resolution of 3.59 Å. Another class of 309,508 particles with CA C-terminus density was selected for non-uniform refinement to the final map with a global resolution of 4.24 Å.

## Atomic model of KAKA/G225R CA hexamer and structural analysis

The model was initially predicted from AlphaFold3[63] and improved in Coot by manually main-chain and sidechain fitting. The refinement of the atomic model was carried out in real space with program Phenix.real_space refine[64], with secondary structure and geometry restrains to prevent overfitting. Structural comparison and visualization were conducted in UCSF ChimeraX[65] and Chimera[66]. All figures of the structures were plotted in UCSF ChimeraX[65] and Chimera.

## KAKA/G225R CA extended C-terminal model building

The initial model for a KAKA/G225R CA trimer of dimer with an extended C-terminal was derived from the KAKA/G225R CA hexamer (PDB 9I8I) using Rosetta[37], by the following procedure: First, three CA hexamers were rigid-body fitted in the density derived for the KAKA/G225R CA tri-hexamer using ChimeraX[65]. Next, the density corresponding to a single monomer at the trimer interface, along with the corresponding rigid-body fitted CA monomer coordinates, was extracted. Residues 1–12 and 220–231 were then removed from the CA monomer, and missing regions were remodeled into the density using RosettaES[39] with the KAKA/G225R CA sequence. In total, 200 models were sequentially generated by building the missing regions one residue at a time and choosing the most favorable conformation through a Monte Carlo procedure[39].

Furthermore, a trimer of dimers assembly was built from the KAKA/G225R CA monomer by applying a C3 symmetric Monte Carlo refinement of the interface sidechain orientation and the distance between monomers. The Monte Carlo refinement was implemented using the Generic Monte Carlo mover in RosettaScripts[38]. The cryoEM density is used as a guiding potential at all steps of the modeling process in Rosetta by using the elec_dens_wt scoring function with a weight of 35. A total of 40 symmetrized and refined models were generated, and the model with the lowest Rosetta score was selected for further study via molecular dynamics.

## Molecular dynamics simulations setup

In preparation for MD simulations of CA trimer of dimers, we prepared models for the KAKA/G225R, KAKA and WT CA as follows: First, we assigned the protonation state of the C-terminal extended KAKA/G225R CA trimer of dimers model at pH 7.0 and added hydrogen atoms as appropriate using the PDB2PQR[67] software version 3.6.1. We then generated models for a KAKA and WT CA trimer of dimers by reverting G225R or all mutations in the KAKA/G225R CA model using the Mutator plugin in the Visual Molecular Dynamics (VMD) software[68]. Afterwards, we solvated each system in a water box using TIP3P[69] water molecules and added Na⁺/Cl⁻ ions around the protein and in bulk using the cionize and autoionize VMD plugins, to a 150 mM NaCl concentration. We then employed the hydrogen mass repartitioning (HMR) procedure[70] to redistribute the mass of heavy atoms bound to hydrogen, allowing us to use a timestep of 4.0 fs during MD simulations for increased sampling efficiency. This procedure resulted in one MD-ready CA trimer of dimers model for each CA sequence. Each MD-ready CA trimer of dimer models consisted of ~268,000 atoms, including solvent, spanning a unit cell of dimensions 173 Å × 173 Å × 88 Å.

## Molecular dynamics simulations

MD-ready systems were minimized and equilibrated by the following procedure: First, each system was minimized using a conjugate gradient algorithm for 50,000 steps. Convergence of the minimization process was verified by ensuring that the energy gradient achieved values below 0.1 kcal mol⁻¹ Å⁻². Next, each system was thermalized by gradually raising the temperature from 50 to 310 K at a rate if 20 K/ns in an NPT ensemble (constant number of particles, pressure $P = 1$ atm, and temperature). During thermalization, the position of protein backbone atoms was harmonically restrained with a spring constant of 10 kcal mol⁻¹ Å⁻². Following thermalization, the positional harmonic restraints are gradually released at a rate of −1 kcal mol⁻¹ Å⁻² every 0.2 ns in an NPT simulation at $T = 310$ K and $P = 1$ atm.

Equilibration MD simulations were then performed for each CA trimer of dimers system for 400 ns in an NPT ensemble at $T = 310$ K and $P = 1$ atm. During equilibration, harmonic positional restraints with a spring constant of 0.5 kcal mol⁻¹ Å⁻² were applied to the carbon alpha atoms of all alpha helices from residues 1 to 218; to maintain the stability of the trimer of dimers in the absence of a continuous CA lattice and to conserve the fitting to the cryoEM density, while linkers and the extended C-terminal segment (residue 220–231) were unrestrained and unbiased. In all simulations, pressure control was applied using a Nose-Hoover Langevin piston with a 200 ps period and a 100 ps decay time, maintaining a constant area in the XY plane of the unit cell while volume was allowed to fluctuate in the Z axis; temperature control was applied using a Langevin thermostat with 0.1 ps⁻¹ damping coefficient. The root mean square deviation (RMSD) of each system stabilized after approximately 50 ns (Supplementary Fig. 8a), and the remaining of 350 ns of the trajectory were utilized as production trajectories for analysis. For each system, equilibration and production simulations were performed in seven independent replicates (summarized in Supplementary Table 5) and analysed separately to ensure the observations were reproducible independent of the initial configuration.

All simulations were performed in the NAMD 3.0.1[71] molecular dynamics engine using the CHARMM36m[72] force field parameters for proteins, water molecules and ions. MD simulations used a 4.0 fs timestep, in accordance with the HMR scheme. Bonds to hydrogen atoms were constrained using the SHAKE[73] and SETTLE[74] algorithms. Nonbonded interactions were calculated with a 12 Å cutoff, 10 Å switching distance, and 14 Å pairlist distance for short-range electrostatics, with long-range electrostatics calculated every 8.0 fs using the particle mesh Ewald summation method[75] with a 1 Å grid spacing. These simulation parameters and force fields are routinely employed in all-atom simulations to study protein-protein interactions in explicit

solvent. The use of HMR provides a twofold increase in computational efficiency while maintaining the accuracy of conventional 2.0 fs time-step simulation for the specified interaction cutoff distances.

After completing the MD simulations, the trajectories were analysed using in-house scripts in VMD[68] to measure contact occupancies, with a threshold of 3.5 Å for salt bridge interactions. Occupancy of the C-terminal segment at the dimer interface was considered when at atoms of the C-terminal region established 3.5 Å contacts with atoms from the neighboring CA monomers' 3–10 helix or helix 9 simultaneously. Convergence of C-terminal segment occupancies at the dimer interface was evaluated based on the variation in occupancy measured over the last 10 ns of the trajectory, with convergence defined as a change in less than 1% (Supplementary Fig. 8b). Volume occupancies were calculated using the volmap tool in VMD[68].

### Reporting summary

Further information on research design is available in the Nature Portfolio Reporting Summary linked to this article.

## Data availability

All data needed to evaluate the conclusions in the paper are present in the paper and/or the Supplementary information, and source data are provided with this paper. The immature and mature HIV-1 capsid lattice structures have been deposited in the Electron Microscopy Data Bank (EMDB) and Protein Data Bank (PDB) under accession codes: EMD-51826 for the K227A immature CA hexamer, EMD-51825 for the KAKA/T8I immature CA hexamer, EMD-51821 for G225R immature CA hexamer, EMD-51822 for the KAKA/G225R immature CA hexamer, EMD-51823 for the KAKA/T8I/G225R immature CA hexamer, EMD-51824 for the A77V/KAKA/T8I/G225R immature CA hexamer, EMD-52724 and PDB 9I8I for the KAKA/G225R mature CA hexamer, EMD-52725 for the KAKA/G225R mature CA tri-hexamer Classs 1, and EMD-52726 for the KAKA/G225R mature CA tri-hexamer Classs 2. All input parameters, structures and example outputs for the MD simulations performed in this study, as well as scripts for analysis of the trajectories are available online in Zenodo (https://doi.org/10.5281/zenodo.16415712). Source data are provided with this paper.

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

## Acknowledgements

We are grateful to Drs Yuewen Sheng and Yun Song for technical support for cryoEM data collection, and to Tapan Kanai for assistance with cryoEM grid preparation. We thank Diamond Light Source for access and support of the cryoEM facilities at the UK National Electron Bio-Imaging Center (eBIC) (proposal NT29812). Computation was performed at the

Oxford Biomedical Research Computing (BMRC) facility, a joint development between the Wellcome Center for Human Genetics (Wellcome Trust Core Award Grant Number 203141/Z/16/Z) and the Big Data Institute (BDI) supported by Health Data Research UK and the NIHR Oxford Biomedical Research Center. This work was supported by the National Institutes of Health (P50AI150481), the UK Wellcome Trust Investigator Award 206422/Z/17/Z (to P.Z.), the UK Wellcome Discovery Award 311427/Z/24/Z (to P.Z.), the UK Biotechnology and Biological Sciences Research Council grant BB/S003339/1 (to P.Z.) and ERC AdG grant 101021133 (to P.Z.). Research in the Freed laboratory is supported by the Intramural Research Program of the Center for Cancer Research, National Cancer Institute, National Institutes of Health. A.B.K. was supported in part by an Intramural AIDS Research Fellowship and the National Institute of Allergy and Infectious Disease (K99AI174891-01 to A.B.K.). Additionally, we would like to acknowledge our collaborative interactions with the Pittsburgh Center for HIV Protein Interactions (U54AI170791). Work in the J.R.P. laboratory was funded in part by the National Institutes of Health through grant R01AI178846 (to J.R.P.). This work used the Delta supercomputing resources at the University of Illinois Urbana-Champaign as well as the Stampede3 supercomputer at the Texas Advanced Computing Center (TACC) through allocation MCB-170096 from the Advanced Cyberinfrastructure Coordination Ecosystem: Services & Support (ACCESS) program, which is supported by National Science Foundation awards #2138259, #2138286, #2138307, #2137603, and #2138296. We acknowledge computational support through the Delaware Advanced Research Workforce and Innovation Network (DARWIN) at the University of Delaware. This research was supported in part by the Intramural Research Program of the National Institutes of Health (NIH). The contributions of the NIH author(s) were made as part of their official duties as NIH federal employees, are in compliance with agency policy requirements, and are considered Works of the United States Government. However, the findings and conclusions presented in this paper are those of the author(s) and do not necessarily reflect the views of the NIH or the U.S. Department of Health and Human Services.

## Author contributions

P.Z., E.O.F. and A.B.K. conceived the research. A.B.K. prepared the immature and mature VLP samples and conducted the forced evolution, infectivity, virus production experiments. J.S. purified WT CA and mutant variants, conducted the in vitro CLP assembly assay. J.S., Y.S. and Y.Z. prepared cryoEM grids and collected cryoEM data. Y.Z. collected cryoET data. Y.Z., L.C., A.L. and N.H. determined the immature Gag structures. Y.Z. carried out the cryoEM single particle analysis, determined the structures and built the model. J.X. conducted the statistical analysis of virion morphology. J.S.R. performed density-guided model building in Rosetta and carried out molecular dynamics (MD) simulations. J.R.P. and J.S.R. analyzed MD simulation results. Y.Z., P.Z., A.B.K., E.O.F., J.S.R. and J.R.P. wrote the paper with the input from all the authors.

## Competing interests

The authors declare no competing interests.
