## [Transparent Peer Review file · Nature Communications]

Structural basis for HIV-1 capsid adaption to a deficiency in IP6 packaging

Corresponding Author: Professor Peijun Zhang

Version 0:

Reviewer comments:

Reviewer #1

(Remarks to the Author)

The manuscript from Zhu et al. describes new structural and functional data on HIV-1 Gag mutants that restore capsid formation in the absence of inositol hexakisphosphate (IP6). In particular, the authors found that G225R, a mutation in the C-terminus of the capsid protein CA, can form capsids independently of IP6. Cryo-EM and MD simulations reveal that the mutated C-terminus adopts a new stabilizing role at the CA hexamer-hexamer interface, making up for the lack of IP6.

This is a very fascinating and thorough look at a specific aspect of HIV-1 capsid assembly. I have a few questions and comments.

Regarding the MD simulations, are the authors confident that their observations are converged? Each system was simulated only once. The three interfaces help with statistics, although the high variability (1.7% - 62.5%) suggest that they are not converged (or may be coupled). If not too difficult, I think another simulation for each system would be helpful to compare.

Figure 5E: Could the authors comment on the role of classification and averaging in terms of the C-terminal density? In other words, are there any classes for which only one or two C-termini were visible? Or does this just manifest as lower densities for each? This is especially relevant as the interactions are described as "transient" (line 364).

Figure 3F: Is the number of tubes and cones the best measure to compare given that the sizes aren't uniform?

Reviewer #2

(Remarks to the Author)

In the manuscript by Yanan Zhu et al., entitled "Structural Basis for HIV-1 Capsid Adaptation to a Deficiency in IP6 Packaging," the authors describe a novel mutation in gag that facilitates the assembly of the HIV capsid in the absence of inositol hexaphosphate (IP6). This study builds upon their previous work, which demonstrated that viruses harboring mutations at both Lys rings (K158A/K227A, referred to as 'KAKA') in the immature Gag lattice can assemble independently of IP6 (Renner N. et al., 2023).

The manuscript employs cryo-ET and subtomogram averaging to analyze the structural integrity of gag VLPs with different mutations, providing an averaged structure of immature gag hexamers with a density corresponding to IP6. The authors identify that the G225R mutation promotes mature capsid assembly under low-IP6 conditions. Additionally, they use cryo-EM, SPA, and MDS to obtain a high-resolution structure of the CA KAKA/G225R hexamer in the presence of IP6.

Overall, the manuscript is well-written, and the experimental data support the conclusions. However, it remains unclear whether the G225R mutation occurs naturally and what its biological relevance might be. Overall, this is a well-executed, descriptive study that contributes to the understanding of HIV capsid assembly.

The authors suggest that their work provides a valuable tool for capsid-related studies and may indicate a previously unrecognized role for the unstructured C-terminus in HIV-1 capsid assembly. However, they do not elaborate on what this potential role might be or how it could be further investigated.

Concerns and suggestions:

1) One suggestion for improvement would be to include the cryo-ET data for G225R in the main figures rather than

relegating it to Supplementary Figure 3.

2) In addition, providing a model summarizing the role of mutations in the assembly of immature gag layer and capsid assembly would be helpful. This would enhance clarity and accessibility for readers.

3) Authors should provide an explanation of why is smaller amount of IP6 is still needed and if it is needed only for the assembly of pentamers. Although KAKA/G225R (Figure 3E) assemble capsids already at 5 μ M IP6, they form long tubes rather than short capsids, indicating that they predominantly assemble hexamers and that for pentamers still IP6 is needed. Authors could quantify separately tubes and cones in F to provide further insights into IP6's role in the assembly of different constructs.

4) How much IP6 is needed to observe in vitro capsid assembly (Figure 3E) in A77V/KAKA/T8I/G225R, and what is the fraction of cones and tubes? One would expect that it assembles cones even at lower IP6 concentrations than KAKA/G225R.

Version 1:

Reviewer comments:

Reviewer #1

(Remarks to the Author)

The authors have done a thorough job in responding to the reviews, especially with the new simulations.

Reviewer #2

(Remarks to the Author)

The authors have addressed all my comments and provided new experimental data, further improving the manuscript. Figure 3 legend has a small typo: "Assembled tubes should be (H) instead of (I)"

This is an interesting work!

Point-by-point responses to the reviewers' comments

Reviewer #1 (Remarks to the Author):

The manuscript from Zhu et al. describes new structural and functional data on HIV-1 Gag mutants that restore capsid formation in the absence of inositol hexakisphosphate (IP6). In particular, the authors found that G225R, a mutation in the C-terminus of the capsid protein CA, can form capsids independently of IP6. Cryo-EM and MD simulations reveal that the mutated C-terminus adopts a new stabilizing role at the CA hexamer-hexamer interface, making up for the lack of IP6.

This is a very fascinating and thorough look at a specific aspect of HIV-1 capsid assembly. I have a few questions and comments.

We appreciate the positive comments from the reviewer.

Regarding the MD simulations, are the authors confident that their observations are converged? Each system was simulated only once. The three interfaces help with statistics, although the high variability (1.7% - 62.5%) suggest that they are not converged (or may be coupled). If not too difficult, I think another simulation for each system would be helpful to compare.

We thank the reviewer for raising this point. To improve the robustness of our observations, we performed six additional independent replicas of the MD simulations for each system (WT, KAKA and KAKA/G225R CA), resulting in a total of 21 independent MD simulations. Across these replicates, we continued to observe high variability in the occupancy of the C-terminus tail at the dimer interface for all CA sequences. Notably, the highest occupancies were consistently observed in the KAKA/G225R CA simulations, with an average occupancy (20.7%) around double what we observe for the KAKA and WT CA simulations. Interestingly, in one of the WT CA simulation replicas, we observed a C-terminal contact event with high residence time. In this event, the C-terminal tail was stabilized by salt-bridge interactions between the CA C-terminal residues R229 and L231 (the terminal residue) with charged residues at the dimer interface, suggesting that it is possible for the WT CA C-terminal tail to interact with the dimer interface. These high residence time interactions are more prevalent in the KAKA G225R CA simulations, as evidenced by the residence time distributions. In the KAKA G225R CA simulations, we observe that G225R transiently forms additional salt bridge interactions with D152 and hydrogen bonds with Q192. These interactions may help stabilize the flexible C-terminus at the dimer interface, contributing to longer contact residence times. We have edited the text and Supplementary Figure 5 to include the data from the additional simulation replicas.

Figure 5E: Could the authors comment on the role of classification and averaging in terms of the C-terminal density? In other words, are there any classes for which only one or two C-termini were visible? Or does this just manifest as lower densities for each? This is especially relevant as the interactions are described as "transient" (line 364).

Thanks for the question. We explored multiple 3D classification strategies to identify classes with one or two C-termini densities without C3 symmetry but were unable to resolve them. This may be due to the dynamic and transient nature of their binding, making it difficult for us to capture sufficient signal for further classification.

Figure 3F: Is the number of tubes and cones the best measure to compare given that the sizes aren't uniform?

The measurement of number of assemblies (tubes and cones, individual, visual) is meant to be complementary to the measurement of assembly by sup-pellet assays (bulk assemble). Given that the sizes of tubes and cores are quite different, they have now been separately quantified (new panels Figure 3F-I).

Reviewer #2 (Remarks to the Author):

In the manuscript by Yanan Zhu et al., entitled “Structural Basis for HIV-1 Capsid Adaptation to a Deficiency in IP6 Packaging,” the authors describe a novel mutation in gag that facilitates the assembly of the HIV capsid in the absence of inositol hexaphosphate (IP6). This study builds upon their previous work, which demonstrated that viruses harboring mutations at both Lys rings (K158A/K227A, referred to as ‘KAKA’) in the immature Gag lattice can assemble independently of IP6 (Renner N. et al., 2023).

The manuscript employs cryo-ET and subtomogram averaging to analyze the structural integrity of gag VLPs with different mutations, providing an averaged structure of immature gag hexamers with a density corresponding to IP6. The authors identify that the G225R mutation promotes mature capsid assembly under low-IP6 conditions. Additionally, they use cryo-EM, SPA, and MDS to obtain a high-resolution structure of the CA KAKA/G225R hexamer in the presence of IP6.

Overall, the manuscript is well-written, and the experimental data support the conclusions.

However, it remains unclear whether the G225R mutation occurs naturally and what its biological relevance might be. Overall, this is a well-executed, descriptive study that contributes to the understanding of HIV capsid assembly.

We thank the reviewer for their positive comments. We have added the following paragraph to the discussion of the manuscript to address this point.

“While the G225R mutation occurs very infrequently in a natural context (observed in 3 of 9494 major subtype sequences analyzed online via the AnalyzeAlign tool provided by Los Alamos National Laboratory (LANL) HIV Sequence Database), it should be noted that in the current study its acquisition occurred through an initial G225S change. G225S is a frequent polymorphism, occurring at a rate of 27.8% in subtype B and 50% of subtype C sequences. G225S has arisen as a compensatory mutation in several different contexts, including as a CTL-escape mutation (Allen et al. JVI. 2005.), and a compensatory mutation restoring replication to the SP1-A3V mutant (Adamson. JVI. 2006), which demonstrates resistance to the maturation inhibitor, PA-457 (later named bevirimat [BVM]). Maturation inhibitors block CA-SP1 processing by stabilizing the 6HB of Gag hexamers in the immature lattice, leading

to defects in capsid assembly during maturation. G225S restores replication to A3V without reversing its defect in CA-SP1 processing. We also recently reported that G225S was acquired upon propagation of CA-K25A (Kleinpeter et al. Nat Comms. 2024), a mutant that is unable to assemble stable capsids. While G225S only mildly restores K25A infectivity, these findings suggest that G225S confers a restorative effect on capsid assembly and stability in non-optimal conditions. G225D was also previously identified as an assembly-deficient mutant whose replication could be restored by the maturation inhibitor PF-46396 (Waki et al. Plos Path. 2012.). This suggests that G225 is critical for proper immature Gag lattice and virus assembly. These previous findings are consistent with our results, which suggest a role for G225 in the assembly of both the immature Gag lattice and capsid.”

The authors suggest that their work provides a valuable tool for capsid-related studies and may indicate a previously unrecognized role for the unstructured C-terminus in HIV-1 capsid assembly. However, they do not elaborate on what this potential role might be or how it could be further investigated.

The G225R mutation suggests that the flexible C-terminus can adopt an ordered conformation that reinforces the hexamer-hexamer interface. Although speculative, we think the role of CA C-terminal fragment maybe parallel to the recently discovered SP2 role in HIV-1 matrix maturation. The unstructured C-terminus may act as a "molecular switch" that facilitates capsid assembly under low-IP6 conditions (such as metabolic constraints). The unstructured C-terminus, long overlooked, could be an important regulatory element in the capsid life cycle. Future comparative studies could be carried out to shed lights on this adaptive mechanism across retroviruses.

Concerns and suggestions:

1) One suggestion for improvement would be to include the cryo-ET data for G225R in the main figures rather than relegating it to Supplementary Figure 3.

We thank the reviewer for the suggestion and have moved the Supplementary Figure3 to **Figure 2**.

2) In addition, providing a model summarizing the role of mutations in the assembly of immature gag layer and capsid assembly would be helpful. This would enhance clarity and accessibility for readers.

We appreciate the reviewer's suggestion and now have included a model summarizing the role of mutations in the assembly of immature and capsid assembly (See Figure R1 below, new Figure 7).

Figure R1 | A schematic model for the role of G225R in HIV-1 mature capsid assembly. (A) The assembly WT mature capsid with high level of IP6 enrichment through the immature Gag assembly. (B) The assembly KAKA/G225R mature capsid with low level of cytosolic IP6, as the mutant is defective in IP6 packaging by immature Gag. The assembly of mutant mature capsid is facilitated by the interactions with the CA C-terminal fragment which is otherwise unstructured and flexible.

3) Authors should provide an explanation of why is smaller amount of IP6 is still needed and if it is needed only for the assembly of pentamers. Although KAKA/G225R (Figure 3E) assemble capsids already at 5 μ M IP6, they form long tubes rather than short capsids, indicating that they predominantly assemble hexamers and that for pentamers still IP6 is needed. Authors could quantify separately tubes and cores in F to provide further insights into IP6's role in the assembly of different constructs.

We appreciated the reviewer's comment. As the reviewer suggested, we have now separately quantified tubes and cones (see Figure R2 below, and new panels Figure 3F-I). The assembly of cones does require higher concentrations of IP6, suggesting that pentamers rely more on IP6. While structured C-terminus strengthens the dimer interface and facilitates the hexameric lattice, IP6 remains important for incorporating CA pentamers.

Figure R2 | The numbers of total assemblies (A) and assembled tubes (B) and cones (C) in each micrograph of WT and mutant CA at indicated IP6 concentrations (n=5).

4) How much IP6 is needed to observe *in vitro* capsid assembly (Figure 3E) in A77V/KAKA/T8I/G225R, and what is the fraction of cones and tubes? One would expect that it assembles cones even at lower IP6 concentrations than KAKA/G225R.

We appreciate the reviewer's comment. We performed *in vitro* assembly assays for the A77V/KAKA/G225R mutant under the same conditions as the other variants. The initial assembly of A77V/KAKA/G225R resembles that of KAKA/G225R and G225R. Interestingly, at high concentrations of IP6 (5mM), A77V/KAKA/G225R conical assemblies are monodispersed, compared to aggregations observed in other variants (see **new panels in Figure3**).

Point-by-point responses to the reviewers' comments

Reviewer #1 (Remarks to the Author):

The authors have done a thorough job in responding to the reviews, especially with the new simulations.

We appreciate the positive comments from the reviewer.

Reviewer #2 (Remarks to the Author):

The authors have addressed all my comments and provided new experimental data, further improving the manuscript. Figure 3 legend has a small typo: "Assembled tubes should be (H) instead of (I)"

This is an interesting work!

We appreciate the positive comments from the reviewer. The typo for Figure 3 legend has been corrected.